# RAD-TFM: Robust and Domain-Adapted Tabular Foundation Models

**Matthew Peroni** [1 2] **Franck Le** [2] **Vadim Sheinin** [2]

## Abstract

Tabular foundation models (TFMs) have shown promise for structured data, but they do not consistently outperform strong traditional ML methods across datasets. Although TFMs can be pretrained entirely on synthetic data, prior work has primarily relied on fixed priors over synthetic data generators. We instead frame generator selection as an adversarial distributional robustness problem: adapt the generator distribution toward datasets with large *optimality gaps*, defined as the shortfall between TFM performance and the best achievable performance on that dataset. We also align synthetic generation with real-world datasets from a target domain, constraining the adversary toward realistic data. These components form the ROBUST AND DOMAIN-ADAPTED TABULAR FOUNDATION MODELS (RAD-TFM) pipeline, a model-agnostic adversarial training framework. Applied to TabPFN V2, RAD-TFM improves performance across six tabular benchmarks, yielding up to an 11% increase in mean normalized AUC over TabPFN V2 and other baselines using only 100k additional synthetic datasets, less than 0.1% of the original pretraining scale.

## 1. Introduction

Recent studies have challenged the deep learning (DL) community to improve upon traditional methods for tabular data, especially boosted trees, which remain strong across many structured prediction tasks (Grinsztajn et al., 2022). In response, substantial progress has been made toward closing this gap through new DL-based tabular methods (Somepalli et al., 2021; Gorishniy et al., 2023; Hegselmann et al., 2023; Hollmann et al., 2023; Carballo et al., 2023; Yan et al., 2024; Zhang & Robinson, 2025). Among these, tabular foundation models (TFMs) based on in-context learning (ICL) have

emerged as a particularly promising direction for classification and regression on structured datasets (Hollmann et al., 2023; Qu et al., 2025; Zhang & Robinson, 2025). TFMs can achieve strong zero-shot performance, often improving upon traditional baselines such as boosted trees (Friedman, 2001; Erickson et al., 2025), while producing predictions on new datasets in milliseconds when GPU-accelerated. However, TFMs still do not uniformly outperform tree-based methods across benchmarks (McElfresh et al., 2024; Ye et al., 2025), motivating methods that better adapt pretraining to the structure of real-world tabular data.

A distinctive feature of current TFMs is that they are pretrained on large collections of synthetic datasets (Hollmann et al., 2023). These datasets are typically sampled from structural causal models (SCMs), whose implicitly parameterized structure provides substantial control over the data generation process (Pearl, 2009; Shanmugam, 2018). Existing publicly available competitive TFMs rely on a fixed prior over SCM parameters (Hollmann et al., 2023; Qu et al., 2025; Zhang & Robinson, 2025). While effective, such fixed priors may underrepresent regions of the generator space that correspond to challenging or practically important real-world datasets. We therefore frame TFM pretraining as an adversarial distributional robustness problem over the SCM parameter space (Madry et al., 2019; Rahimian & Mehrotra, 2019): instead of sampling from a static generator prior, we adapt the generator distribution toward datasets on which the current TFM underperforms. Related work has explored a narrower GAN-style approach that adjusts the weights of a specific class of SCMs during TFM training (Wu & Bergman, 2025); in contrast, we develop a broader DRO framework for robust and domain-adapted TFM pretraining. We formalize model underperformance using an *optimality gap*, which measures the gap between the TFM's performance and an estimate of the best achievable performance on a synthetic dataset.

Beyond robustness, we also seek to make adversarial training realistic and domain-adapted. Recent work such as Real-TabPFN has shown that incorporating real-world data into TFM training can improve performance (Garg et al., 2025). However, finetuning directly on real datasets can be limited by the scarcity of large, high-quality tabular datasets, especially in specialized domains such as genomics or the physical sciences. Rather than train only on real data, we

[1]MIT [2]IBM Research. Correspondence to: Matthew Peroni
<mperoni1@mit.edu>.

*Proceedings of the 43rd International Conference on Machine Learning*, Seoul, South Korea. PMLR 306, 2026. Copyright 2026 by the author(s).

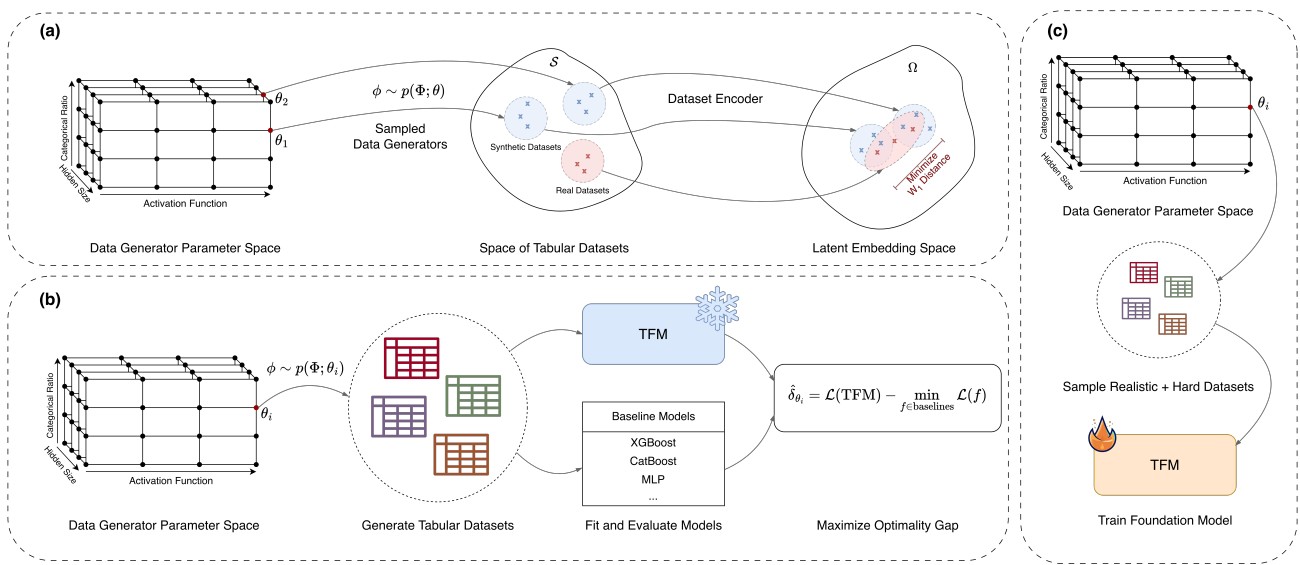

*Figure 1.* Overview of RAD-TFM. In part **(a)**, we learn a mixture distribution over synthetic data generators that aligns with real-world datasets along three dimensions: *dataset schema, geometry, and learnability*. We then solve the core distributionally robust optimization problem by alternating between two stages: **(b)** searching over the SCM parameter space, we find synthetic data distributions with large estimated optimality gaps, and **(c)** sampling parameters proportional to their optimality gap and proximity to the learned reference distribution, we generate synthetic datasets and continue training the TFM.

constrain synthetic generation to match real-world datasets from a target domain. Prior work on tabular synthetic data generation often learns a generator that mimics a single real dataset, frequently using GAN-based approaches (Choi et al., 2018; Park et al., 2018; Xu et al., 2019). In contrast, we construct a *distribution over SCM-based dataset generators* that aligns, in aggregate, with real-world datasets across meta-features capturing schema, geometry, and learnability. Together, these components form ROBUST AND DOMAIN-ADAPTED TABULAR FOUNDATION MODELS (RAD-TFM), a model-agnostic two-stage adversarial training pipeline for TFMs. Applied to TabPFN V2 (Hollmann et al., 2025), RAD-TFM uses only 100k additional synthetic training datasets, less than $0.1\%$ of the original pretraining scale, to substantially improve performance across general and application-specific tabular benchmarks.

## 2. Problem Definition

We first formalize the generator space used for TFM pretraining. Let $\Phi$ denote a family of data-generating mechanisms, where each $\phi \in \Phi$ induces a distribution over datasets $D \sim p(\phi)$. In this work, $\phi$ is an SCM implemented as a randomly initialized MLP, following Hollmann et al. (2023); full details are given in Appendix D. Each SCM is sampled from a parameterized distribution over generator hyperparameters, such as depth, hidden dimension, activation function, and categorical feature ratio. We write this as $\phi \sim p(\Phi; \theta)$, where $\theta \in \Theta$ parameterizes the distribution over SCMs.

Let $g_{\mathbf{W}}$ be a PFN-style predictive model with parameters $\mathbf{W}$. Given labeled context samples $D$ and an unlabeled test point $x_t$, the model predicts $y_t$ in-context. For a generator $\phi$, we define the PFN loss as

$$\mathcal{L}_{PFN}(\mathbf{W}; \phi) = \mathbb{E}_{\substack{(\{(x_t, y_t)\} \cup D) \\ \sim p(\phi)}} \Big[ -\log g_{\mathbf{W}}(y_t \mid x_t, D) \Big]. \tag{1}$$

Standard PFN pretraining minimizes the expected loss over generators sampled from a fixed prior:

$$\min_{\mathbf{W}} \mathbb{E}_{\phi \sim p(\Phi; \theta)} \left[ \mathcal{L}_{PFN}(\mathbf{W}; \phi) \right]. \tag{2}$$

Our goal is to replace this static prior with an adversarial generator distribution that emphasizes regions where the current TFM can improve. Maximizing loss alone is insufficient, since some generators produce datasets that are intrinsically hard for any model. We therefore define an *optimality gap*: for a generator $\phi$, let $(Z_x, Z_y) \sim p(\phi)$. The Bayes-optimal cross-entropy loss is $H_\phi(Z_y \mid Z_x)$, so

$$\mathcal{L}_{PFN}(\mathbf{W}; \phi) - H_\phi(Z_y \mid Z_x) \geq 0 \tag{3}$$

measures the gap between the TFM and the best achievable predictor. This yields the ideal adversarial objective

$$\min_{\mathbf{W}} \max_{\theta \in \Theta} \mathbb{E}_{\phi \sim p(\Phi; \theta)} \left[ \mathcal{L}_{PFN}(\mathbf{W}; \phi) - H_\phi(Z_y \mid Z_x) \right]. \tag{4}$$

Since $H_\phi(Z_y \mid Z_x)$ is generally unavailable, we approximate the Bayes-optimal loss using strong baseline estimators $f_1, \ldots, f_e$, such as XGBoost, CatBoost, and Random

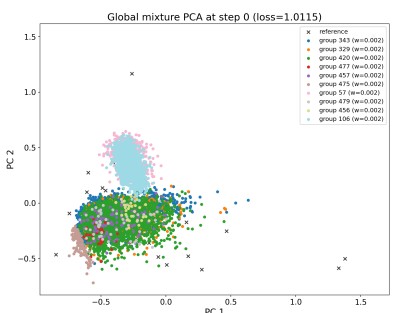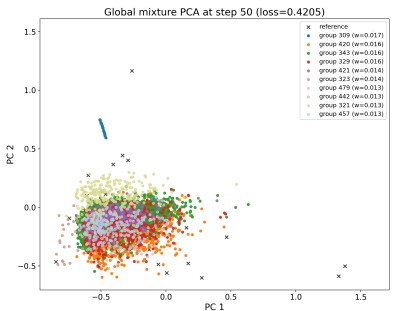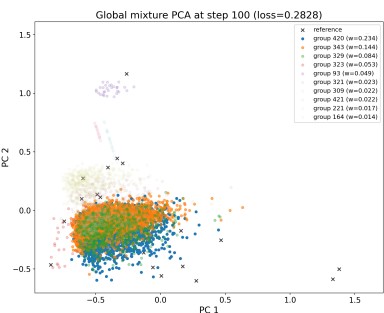

*Figure 2.* Our domain-adaptation methodology applied to the engineering datasets from OpenML (Bischl et al., 2021). The black marks represent the embeddings of the real datasets, and each color corresponds to a different parameterization of the data generator distribution. We learn a mixture distribution over these parameters that minimizes the $W_1$ distance to the real data distribution.

Forests (Chen & Guestrin, 2016; Dorogush et al., 2018; Breiman, 2001). We define the empirical optimality gap

$$\widehat{\delta}_\theta(\mathbf{W}) = \mathbb{E}_{\phi \sim p(\Phi;\theta)} \left[ \mathcal{L}_{PFN}(\mathbf{W}; \phi) - \min_{i \in [e]} \mathcal{L}_{PFN}(f_i; \phi) \right]. \tag{5}$$

Because $\min_i \mathcal{L}_{PFN}(f_i; \phi) \geq H_\phi(Z_y \mid Z_x)$, this gives a lower bound on the true optimality gap; in particular, $\widehat{\delta}_\theta(\mathbf{W}) > 0$ implies that the TFM underperforms at least one baseline on generators sampled from $\theta$.

Finally, to avoid overfitting to a single generator parameterization and to support domain adaptation, we formulate adversarial training as a distributionally robust optimization problem. Let $\Theta$ be discretized and let $Q \in \Delta_\Theta$ be the adversary's distribution over generator parameters. Given a reference distribution $P \in \Delta_\Theta$ aligned with a target domain, we constrain the adversary by a KL ball:

$$\min_{\mathbf{W}} \max_{\substack{Q \in \Delta_\Theta \\ D_{KL}(Q\|P) \leq \rho}} \mathbb{E}_{\theta \sim Q} \left[ \widehat{\delta}_\theta(\mathbf{W}) \right]. \tag{6}$$

As shown in Appendix C, the optimal adversarial distribution satisfies

$$q_i^* \propto p_i \exp\left( \eta \cdot \widehat{\delta}_{\theta_i}(\mathbf{W}) \right), \tag{7}$$

where $\eta$ is determined by the KL radius $\rho$, reference distribution $P$, and estimated gaps. Thus, adversarial training reweights generators according to both domain relevance under $P$ and current model weakness under $\widehat{\delta}_{\theta_i}(\mathbf{W})$.

## 3. RAD-TFM: Domain-Adapted Robust Training for TFMs

RAD-TFM approximately solves Equation 6 in two stages: first estimating a domain-specific reference distribution $P$ over SCM generators, then alternating between adversarial parameter search and continued TFM training.

**Learning a domain-aligned reference distribution.** To connect heterogeneous real-world tabular datasets to the SCM parameter space $\Theta$, we embed each dataset into a shared meta-feature space. Following standard meta-learning approaches (Brazdil et al., 2009; Pinto et al., 2016; Rivolli et al., 2019), our encoder $f : \mathcal{S} \to \mathbb{R}^k$ summarizes dataset schema, geometry, and learnability; details are in Appendix A. Given real datasets $D_r = \{(\mathbf{X}_i, \mathbf{y}_i)\}_{i=1}^{n_r}$ from a target domain, we view their embeddings as samples from a distribution $P_r$. Each generator parameter $\theta_i$ induces a distribution $Q(\theta_i)$ over synthetic dataset embeddings, and we learn a generator mixture by solving

$$P^* \in \arg\min_{P \in \Delta_\Theta} W_1 \left( P_r, \sum_{i=1}^M p_i Q(\theta_i) \right). \tag{8}$$

We optimize the entropy-regularized OT distance via Sinkhorn iterations (Cuturi, 2013). Since $\Theta$ is large, we use Bayesian optimization (Frazier, 2018) to find generator distributions that partially cover $P_r$, then solve Equation 8 over the discovered candidates. The result is a sparse reference distribution $P$ that aligns synthetic generation with the target domain. Figure 2 illustrates this process for the Engineering benchmark, where the learned generator mixture progressively covers the real-dataset embeddings in the meta-feature space.

**Adversarial robust training.** Given $P$, RAD-TFM alternates between maximization and minimization stages. In the maximization stage, we freeze the pretrained TFM $g_\mathbf{W}$ and use black-box optimization (Akiba et al., 2019; Watanabe, 2023; Bergstra et al., 2011) to search over $\Theta$. For each proposed $\theta_i$, we sample synthetic datasets and estimate $\widehat{\delta}_{\theta_i}(\mathbf{W})$ by comparing the TFM loss to the best loss among baseline estimators $f_1, \ldots, f_e$. This yields candidate parameters and gap estimates $\{(\theta_i, \widehat{\delta}_{\theta_i})\}_{i=1}^{n_{trials}}$.

In the minimization stage, we sample generator parameters according to the optimal adversarial distribution from

*Table 1.* Experiment results across three tabular dataset benchmarks. We report normalized AUC scaled to [0,1] for each dataset (McElfresh et al., 2024; Hollmann et al., 2025); $^*$ denotes $p < 0.05$ in Wilcoxon signed-rank tests against RAD-TFM.

| Benchmark | Metric | LR | MLP | RF | CatBoost | XGB | TabPFN$_{n.e.}$ | TabPFN$_{n.e.}$ (FT) | TabPFN$_{n.e.}$ (RAD-TFM) |
|---|---|---|---|---|---|---|---|---|---|
| **TabPertNet**(Ye et al., 2024) | Mean rank AUC | 5.4 | 6.0 | 4.6 | 4.2 | 5.2 | 3.7 | 3.9 | **3.1** |
| | Mean Norm. AUC | 0.4684 | 0.3627 | 0.6646 | 0.7404 | 0.5743 | 0.7306 | 0.7299 | **0.8486**$^*$ |
| | Rank-1 Wins | 6 | 5 | 7 | 6 | 7 | 9 | 2 | **13** |
| **TabArena**(Erickson et al., 2025) | Mean rank AUC | 5.7 | 7.2 | 5.5 | 4.0 | 5.2 | 3.6 | 2.7 | **2.1** |
| | Mean Norm. AUC | 0.5070 | 0.2513 | 0.6870 | 0.8214 | 0.7293 | 0.8829 | 0.9215 | **0.9421**$^*$ |
| | Rank-1 Wins | 2 | 0 | 0 | 1 | 3 | 1 | 4 | **9** |
| **Engineering** (Bischl et al., 2021) | Mean rank AUC | 6.9 | 7.3 | 4.3 | 3.4 | 4.9 | 3.3 | 3.4 | **2.6** |
| | Mean Norm. AUC | 0.2944 | 0.3951 | 0.8405 | 0.8952 | 0.7805 | 0.9004 | 0.8998 | **0.9313**$^*$ |
| | Rank-1 Wins | 0 | 0 | 1 | 4 | 2 | 0 | 0 | **7** |

Equation 6, where the sampling weight increases with both the reference mass $p_i$ and the estimated gap $\widehat{\delta}_{\theta_i}(\mathbf{W})$. We then continue PFN training on datasets generated from these sampled parameters. After a fixed number of training steps, we repeat the maximization stage with the updated TFM. To reduce unlearning, we periodically include the original pre-trained TFM as an additional baseline in the gap estimate.

## 4. Experiments

We evaluate RAD-TFM by initializing from the open-source TabPFN V2 classifier (Hollmann et al., 2025) and applying our robust training pipeline to TabPertNet (Ye et al., 2024), TabArena (Erickson et al., 2025), and an Engineering benchmark constructed from OpenML datasets (Bischl et al., 2021). Additional experiments on PMLB, Genomics, and Healthcare, along with full experimental details and ablations, are provided in Appendix E. We compare against Random Forest, CatBoost, XGBoost, Logistic Regression, MLPs, TabPFN V2, and TabPFN V2 finetuned for 10 epochs. Following prior work, we report normalized ROC-AUC, mean rank, and rank-1 wins (McElfresh et al., 2024; Hollmann et al., 2025; Garg et al., 2025). Across benchmarks, RAD-TFM uses only 100k additional synthetic datasets, making the continued-training budget small relative to the original TabPFN V2 pretraining scale.

Across all three reported benchmarks, RAD-TFM achieves the best mean rank, mean normalized AUC, and number of rank-1 wins among all evaluated methods. The largest gain occurs on TabPertNet, where mean normalized AUC increases from 0.7306 for TabPFN V2 to 0.8486 with RAD-TFM. On TabArena and Engineering, RAD-TFM also improves over both the original and finetuned TabPFN baselines, showing that robust synthetic training improves both broad benchmark and domain-specific performance. To assess significance, we first apply the Friedman test across model classes, then use pairwise Wilcoxon signed-rank tests comparing RAD-TFM against each alternative model (Demšar, 2006; Benavoli et al., 2016; Conover, 1999). For all three reported benchmarks, the Friedman tests are sig-

nificant and RAD-TFM improves over each baseline under Wilcoxon tests at $p < 0.05$.

Additional analyses in Appendix E further distinguish RAD-TFM from standard finetuning. Across benchmarks, RAD-TFM produces more outright wins and more *leaps*, defined as datasets where the adapted TabPFN surpasses at least one non-TabPFN baseline that the original model did not outperform. The learned adversarial distributions also select substantially different high-weight SCM parameters across benchmarks, including feature count, categorical feature ratio, missingness, activation, sparsity, and noise. Thus, RAD-TFM does not merely upweight harder synthetic datasets; it learns domain-specific generation regimes needed to cover different real-world benchmark distributions.

## 5. Conclusion

We introduce RAD-TFM, a model-agnostic framework for robust, domain-adaptive continued training of TFMs. By combining DRO with controllable SCM-based synthetic generation, RAD-TFM targets realistic generator regimes where the current TFM underperforms, improving TabPFN V2 across six benchmarks using only 100k additional synthetic datasets, less than $0.1\%$ of the original pretraining scale. While our experiments focus on TabPFN V2 classification, the framework naturally extends to other TFMs, including Mitra and TabICL, and to regression tasks. Future work includes expanding the SCM parameter space, learning richer dataset embeddings for domain alignment, and studying how robust generator reweighting interacts with larger-scale TFM pretraining. These results suggest that robust synthetic training can improve TFMs beyond standard finetuning by adapting the generator distribution, rather than updating model weights on limited real data alone.

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

## A. Details for Domain-Adapted TFMs

In this Section, we provide a detailed overview of the domain-adaptation process introduced in Section 3. We begin by defining the set of meta-features used to encode tabular datasets, then define our surrogate reward for the Bayesian Optimization process to maximize partial coverage of the real data distribution, and finally introduce our algorithm for constructing the global mixture over synthetic data generators. Although the overall pipeline involves multiple steps, the meta-features we define are inexpensive to compute, and we are able to solve the global mixture optimization efficiently via the Sinkhorn algorithm. As a result, we found that for all of our experiments, we can recover the approximate reference distribution over synthetic data generators in under 10 minutes, using a single A100 GPU on standard hardware. Further, this process only needs to be run once for each domain, since the same reference distribution can be re-used across many training runs of the RAD-TFM algorithm. We summarize the pseudocode for our domain-adaptation pipline in Algorithms 1 and 2.

**Meta-Feature Extraction.** For a tabular dataset $(X, y)$ with $n$ samples and $p$ input features, we construct a meta-feature embedding

$$\phi(X, y) = \begin{bmatrix} \phi_{\mathrm{schema}}(X) \\ \phi_{\mathrm{geom}}(X) \\ \phi_{\mathrm{learn}}(X, y) \end{bmatrix} \in \mathbb{R}^d.$$

The representation is designed to separate three sources of variation: coarse dataset schema and marginals, affine-invariant geometric structure of the numeric covariates, and inexpensive proxies for supervised learnability. In the classification setting used in our experiments, the corresponding block dimensions are $(d_{\mathrm{schema}}, d_{\mathrm{geom}}, d_{\mathrm{learn}}) = (24, 24, 16)$.

Throughout, for any finite vector $v = (v_1, \ldots, v_m)$, we write

$$S(v) = \big(\mathrm{mean}(v),\ \mathrm{std}(v),\ Q_{0.25}(v),\ \mathrm{median}(v),\ Q_{0.75}(v)\big)$$

for the five-dimensional summary operator applied repeatedly below. The schema block contains the global descriptors: $\log(1 + p)$, categorical feature ratio, total missing fraction, together with the fraction of categorical features that are binary, and the summaries $S(\text{per-feature missing rate})$, $S(\text{numeric unique ratio})$, $S(\text{categorical unique ratio})$, $S(\text{categorical normalized entropy})$. Note that for this work, we do not encode schema information regarding the number of samples in the dataset, since we currently fix the number of samples generated for synthetic datasets to be a constant during model training. In future work, we plan to include these features of the dataset schema, but we exclude them for now for training stability purposes. These quantities preserve coarse structural properties of the dataset that should not be removed by downstream normalization.

Let $X_{\mathrm{num}} \in \mathbb{R}^{n \times p_{\mathrm{num}}}$ denote the numeric submatrix. For each numeric feature $j$, we compute the mean $\mu_j$ and standard deviation $\sigma_j$ using observed entries only, impute missing values by $\mu_j$, and form the standardized column

$$\widetilde{X}_{ij} = \frac{X_{ij} - \mu_j}{\sigma_j}.$$

All geometric features are computed from $\widetilde{X}_{\mathrm{num}}$, making this block invariant to per-feature affine transformations $X_{ij} \mapsto a_j X_{ij} + b_j$ with $a_j > 0$. The geometric block contains $S(|\text{feature skewness}|)$, $S(\text{feature kurtosis})$, and $S(\text{feature outlier rate})$, where the outlier rate is the fraction of observed standardized entries satisfying $|\widetilde{X}_{ij}| > 3$. To capture dependence structure, we form the correlation matrix $R$ of the standardized numeric features and include

$$S\big(|R_{jk}| : j < k\big), \qquad \max_{j<k} |R_{jk}|,$$

together with three spectral summaries,

$$\frac{\lambda_1}{\sum_{\ell} \lambda_{\ell}}, \qquad \frac{\sum_{\ell=1}^{\min(5, r)} \lambda_{\ell}}{\sum_{\ell} \lambda_{\ell}}, \qquad \exp\Big(-\sum_{\ell} \pi_{\ell} \log \pi_{\ell}\Big),$$

where $\lambda_1 \geq \lambda_2 \geq \cdots \geq 0$ are the eigenvalues of $R$, $r$ is the rank, and $\pi_{\ell} = \lambda_{\ell} / \sum_k \lambda_k$. The final term is the entropy-based effective rank.

The learnability block augments these unsupervised statistics with inexpensive surrogates for task difficulty. Specifically, it includes #classes, $H(Y)$, $\max_c \Pr(Y = c)$, $\min_c \Pr(Y = c)$, where $H(Y) = -\sum_c \Pr(Y = c) \log \Pr(Y = c)$. It further

includes the summaries $S(\eta_j^2)$ over numeric features, and $S(V_j)$ over categorical features, where $\eta_j^2$ is the ANOVA effect size between numeric feature $j$ and the class label, and $V_j$ is Cramér's $V$ between categorical feature $j$ and the class label. Finally, we include two global proxies for separability: leave-one-out nearest-centroid accuracy, and the ratio

$$\frac{\text{mean pairwise distance between class centroids}}{\text{mean within-class distance}}.$$

Given a collection of reference datasets $\{(X^{(i)}, y^{(i)})\}_{i=1}^N$, we first compute raw embeddings

$$u^{(i)} = \phi(X^{(i)}, y^{(i)}) \in \mathbb{R}^d.$$

For each coordinate $k$, we then estimate the reference-set mean and variance,

$$\mu_k = \frac{1}{N} \sum_{i=1}^N u_k^{(i)}, \qquad s_k^2 = \frac{1}{N} \sum_{i=1}^N (u_k^{(i)} - \mu_k)^2,$$

and standardize each embedding coordinate according to

$$z_k^{(i)} = \frac{u_k^{(i)} - \mu_k}{s_k}.$$

Any generated dataset is mapped into the same latent space using the *same* $(\mu_k, s_k)$ estimated from the reference collection. After this per-coordinate standardization, we apply an explicit block reweighting. Writing

$$z = \begin{bmatrix} z_{\text{schema}} \\ z_{\text{geom}} \\ z_{\text{learn}} \end{bmatrix}, \qquad z_b \in \mathbb{R}^{d_b},$$

we define the final transformed representation by

$$\widehat{z_b} = \sqrt{\frac{w_b}{d_b}}\, z_b,$$

where $w_b > 0$ is a user-specified block weight. Distances are then evaluated in the transformed space:

$$\|\widehat{z} - \widehat{z}'\|_2^2 = \sum_b \frac{w_b}{d_b} \|z_b - z_b'\|_2^2.$$

This normalization prevents large blocks from dominating the geometry solely by virtue of dimensionality. If each standardized coordinate has variance approximately one and within-block correlations are not too strong, then

$$\mathbb{E}\big[\|z_b\|_2^2\big] \approx d_b, \qquad \mathbb{E}\big[\|\widehat{z_b}\|_2^2\big] = \frac{w_b}{d_b} \mathbb{E}\big[\|z_b\|_2^2\big] \approx w_b.$$

Thus the expected squared contribution of block $b$ is controlled by $w_b$ rather than by its raw dimension. In practice, we set $w_b = \frac{1}{3}$ for each of the three categories, balancing the contribution of each to the distance metric.

**Parameter Search for Partial Coverage.** With our dataset encoder and distance metric defined, we now introduce our search method for finding parameterizations of synthetic data generators that align with real-world data from some reference corpus. Since this is an unconstrained black-box optimization problem over a discrete search space, we use Bayesian Optimization to find parameters that cover parts of the reference corpus well (Frazier, 2018). To accomplish this, we now define our surrogate objective function that rewards partial coverage of the embedding space spanned by the reference corpus. For a given reference corpus of datasets, let $\{\mathbf{x}_i\}_{i=1}^n \subset \mathbb{R}^d$ denote its standardized reference embeddings after the same preprocessing and block reweighting described in the previous section. A trial parameter vector $\theta$ induces a *set* of synthetic datasets; in the implementation, we generate $m$ datasets from $\phi \sim p(\Phi; \theta)$, embed them, and obtain a cloud $\{\mathbf{x}_j(\theta)\}_{j=1}^m \subset \mathbb{R}^d$. The search objective is then defined in terms of how well this cloud covers the reference embeddings.

For each reference point $\mathbf{x}_i$, the we compute a soft nearest-neighbor distance to the generated cloud,

$$d_i(\theta) = -\tau \log \left( \frac{1}{m} \sum_{j=1}^{m} \exp \left( -\frac{\|\mathbf{x}_i - \mathbf{x}_j(\theta)\|_2}{\tau} \right) \right), \tag{9}$$

with temperature $\tau > 0$. Distances are Euclidean and $\tau = 1$ by default. As $\tau \to 0$, $d_i(\theta)$ approaches $\min_j \|\mathbf{x}_i - \mathbf{x}_j(\theta)\|_2$, while for larger $\tau$ it becomes a smoother average over nearby generated embeddings. This gives a differentiable surrogate for pointwise coverage without requiring an exact minimum.

The scalar objective returned to the optimizer at iteration $t$ is a residual-weighted average of these pointwise distances,

$$f_t(\theta) = -\sum_{i=1}^{n} r_i^{(t)} d_i(\theta), \qquad r_i^{(t)} \geq 0, \qquad \sum_{i=1}^{n} r_i^{(t)} = 1. \tag{10}$$

Maximizing $f_t$ is therefore equivalent to minimizing a weighted coverage loss. The weights $r_i^{(t)}$, which we refer to as the *residuals*, capture which parts of the reference data remain poorly covered by the synthetic data generators found so far. To define these residual weights, we track the coverage distance $c_i^{(t)}$ for each reference point. After evaluating a new parameter vector $\theta_t$, the candidate distances $d_i(\theta_t)$ are merged with the existing coverage state by a hard minimum,

$$c_i^{(t+1)} = \begin{cases} d_i(\theta_t), & t = 0, \\ \min\left(c_i^{(t)}, d_i(\theta_t)\right), & t \geq 1. \end{cases} \tag{11}$$

Thus $c_i^{(t)}$ records the best coverage attained so far by any previously evaluated generator in that cluster. The residual weights for the next trial are then recomputed from the updated coverage distances. The update is the linear normalization

$$r_i^{(t+1)} = \frac{c_i^{(t+1)}}{\sum_{\ell=1}^{n} c_\ell^{(t+1)}}. \tag{12}$$

Reference points that are already well covered have small $c_i^{(t+1)}$ and therefore receive little residual mass, whereas persistently under-covered points retain large weights. The initial residual state is uniform, $r_i^{(0)} = 1/n$. Consequently, early trials try to improve average coverage of the whole space, while later trials are pushed toward the portions of embedding space that remain inadequately represented. This is why we refer to the score as a *partial coverage* surrogate, since a single generator is not required to match the entire reference corpus by itself. Instead, the sequential parameter search encourages successive generators to cover complementary regions of the target embedding distribution. In practice, as we can see in Figure 2, there are no generators the adequately cover the entire space spanned by the reference corpus. Instead, by finding generators that partially cover this space and combining them, we are able to sufficiently cover the reference corpus.

**Solving the Global Mixture Problem.** After the parameter search from the previous section is performed, we obtain $T$ candidate generators. Generator $t$ is represented by a cloud of embedded synthetic datasets $\{\boldsymbol{x}_j(\theta_t)\}_{j=1}^{m_t} \subset \mathbb{R}^d$, while the target collection of real datasets is represented by reference embeddings $\{\boldsymbol{x}_i\}_{i=1}^{n} \subset \mathbb{R}^d$. In practice, we set $m_t = m$ as a constant for consistency across trials. We want to fit a global distribution over generators by choosing mixture weights $\boldsymbol{\alpha} = (\alpha_1, \ldots, \alpha_T)$ on the simplex that minimizes the transport distance between this global distribution over synthetic data generators and the distribution over the reference datasets.

The reference embeddings define the empirical measure

$$\mu = \frac{1}{n} \sum_{i=1}^{n} \delta(\boldsymbol{x}_i), \tag{13}$$

and a candidate generator mixture induces

$$\nu_{\boldsymbol{\alpha}} = \sum_{k=1}^{T} \alpha_t \left( \frac{1}{m_t} \sum_{j=1}^{m_t} \delta(\boldsymbol{x}_j(\theta_t)) \right). \tag{14}$$

Thus each synthetic table embedding produced by generator $k$ receives mass $\alpha_t/m_t$. We compare $\mu$ and $\nu_{\boldsymbol{\alpha}}$ using squared Euclidean cost $c(\boldsymbol{x}, \boldsymbol{y}) = \|\boldsymbol{x} - \boldsymbol{y}\|_2^2$ and the entropically regularized optimal transport objective

$$\mathrm{OT}_{\varepsilon}(\mu, \nu_{\boldsymbol{\alpha}}) = \min_{\boldsymbol{\Pi} \in U(\boldsymbol{a}, \boldsymbol{b}(\boldsymbol{\alpha}))} \langle \boldsymbol{\Pi}, \boldsymbol{C} \rangle + \varepsilon \sum_{i,j} \Pi_{ij}(\log \Pi_{ij} - 1), \tag{15}$$

where $\varepsilon > 0$ is the regularization parameter, $\boldsymbol{a}_i = 1/n$, $\boldsymbol{b}(\boldsymbol{\alpha})$ collects the per-sample masses $\alpha_t/m_t$, and $U(\boldsymbol{a}, \boldsymbol{b})$ is the set of couplings with marginals $\boldsymbol{a}$ and $\boldsymbol{b}$. The entropic term makes the problem strictly convex in the transport plan and enables efficient solution by Sinkhorn matrix scaling (Cuturi, 2013).

Rather than solving the transport problem from scratch by general-purpose linear programming, we use the Sinkhorn algorithm in the log domain. Writing $C_{ij} = \|\boldsymbol{x}_i - \boldsymbol{y}_j\|_2^2$, the dual potentials are updated as

$$f_i \leftarrow \varepsilon \left( \log a_i - \log \sum_j \exp\left( \frac{-C_{ij} + g_j}{\varepsilon} \right) \right), \tag{16}$$

$$g_j \leftarrow \varepsilon \left( \log b_j - \log \sum_i \exp\left( \frac{-C_{ij} + f_i}{\varepsilon} \right) \right), \tag{17}$$

which yields the transport plan $\Pi_{ij} = \exp((-C_{ij} + f_i + g_j)/\varepsilon)$. This log-domain form is numerically stable and gives an efficient inner solver for repeated OT evaluations.

---

**Algorithm 1** Pseudocode for Domain-Adapted TFMs

---

**Require:** Reference datasets $\mathcal{D}_{\mathrm{ref}} = \{(\boldsymbol{X}^{(i)}, \boldsymbol{y}^{(i)})\}_{i=1}^N$, parameter space $\Theta$, trials $T$, datasets per trial $m$, soft-coverage temperature $\tau$, transport regularization $\varepsilon$, Sinkhorn iterations $L$, outer iterations $O$
 1: Compute reference embeddings $\{\boldsymbol{x}_i\}_{i=1}^N$ from $\mathcal{D}_{\mathrm{ref}}$
 2: Standardize and reweight the embedding coordinates using statistics estimated from $\mathcal{D}_{\mathrm{ref}}$
 3: Initialize an empty list of candidate generators and generated embedding clouds
 4: Initialize residual weights uniformly as $\boldsymbol{r}^{(0)} = \frac{1}{N}$
 5: $\{(\theta_t, \{\mathbf{x}_j(\theta_t)\}_{j=1}^m)\}_{t=1}^T \leftarrow \mathrm{DomainParamSearch}(\{\boldsymbol{x}_i\}_{i=1}^N, \mathbf{r}, \Theta, m, \tau, T)$
 6: Form the empirical reference measure $\mu$ from $\{\boldsymbol{x}_i\}_{i=1}^N$
 7: For each candidate generator $\theta_t$, form the empirical measure supported on its generated cloud $\mathcal{Y}_t = \frac{1}{m} \sum_{j=1}^m \delta(\boldsymbol{x}_j(\theta_t))$
 8: Define $\nu_{\boldsymbol{\alpha}} = \sum_{t=1}^T \alpha_t \mathcal{Y}_t$ as the global mixture over synthetic data generators
 9: Solve $\boldsymbol{\alpha}^\star \in \arg\min_{\boldsymbol{\alpha} \in \Delta^{M-1}} \mathrm{OT}_{\varepsilon}(\mu, \nu_{\boldsymbol{\alpha}})$ using gradient-based optimization over mixture logits, with $\mathrm{Sinkhorn}(\mu, \nu_{\boldsymbol{\alpha}}, \varepsilon, L)$ as the inner OT solver, for $O$ outer iterations
10: **return** Global mixture over generator parameters $\boldsymbol{\alpha}^\star$

---

We then solve the outer mixture problem by minimizing the regularized transport cost with respect to $\boldsymbol{\alpha}$:

$$\min_{\boldsymbol{\alpha} \in \Delta^{T-1}} \mathrm{OT}_{\varepsilon}(\mu, \nu_{\boldsymbol{\alpha}}). \tag{18}$$

In practice, we reparameterize $\boldsymbol{\alpha}$ by unconstrained logits $\boldsymbol{\eta} \in \mathbb{R}^T$ with $\alpha_t = \exp(\eta_t)/\sum_{\ell=1}^T \exp(\eta_\ell)$, and optimize $\boldsymbol{\eta}$ by gradient descent. Each outer step constructs the sample-weight vector $\boldsymbol{b}(\boldsymbol{\alpha})$, calls the Sinkhorn solver to evaluate $\mathrm{OT}_{\varepsilon}(\mu, \nu_{\boldsymbol{\alpha}})$, and backpropagates through that solution to update the logits. In this way, Sinkhorn serves as the inner OT subroutine inside the global mixture optimization. The result is a set of mixture weights over generators whose aggregated embedding distribution best matches the reference embeddings under the entropically regularized transport geometry.

## B. RAD-TFM Algorithm Details

We summarize ROBUST AND DOMAIN-ADAPTED TABULAR FOUNDATION MODELS in Algorithms 3 and 4. A description of the algorithm is given in Section 3. To begin, we perform the domain-adaption search from given by Algorithm 1 to construct a reference distribution $P$ over SCM parameters that aligns the synthetic data with a given reference corpus of real data. We then alternate between a maximization stage, in which we search over SCM parameters to find those with large

---

**Algorithm 2** DOMAINPARAMSEARCH

---

**Require:** Reference embeddings $\{x_i\}_{i=1}^N$, initial residuals $r^{(0)}$, parameter space $\Theta$, datasets per trial $m$, soft-coverage temperature $\tau$, trials $T$
1: Initialize cover distances $c^{(0)} = (+\infty, \ldots, +\infty)$
2: Initialize history $\mathcal{H}^{(0)} = \emptyset$
3: **for** $t = 1, \ldots, T$ **do**
4:     $\theta_t \leftarrow \text{BayesOPT}(\mathcal{H}^{(t-1)}, \Theta)$
5:     Generate $m$ synthetic datasets from $\theta_t$, compute their embeddings, and collect them as $\{\mathbf{x}_j(\theta_t)\}_{j=1}^m$
6:     Compute the pointwise soft coverage distances $\{d_i(\theta_t)\}_{i=1}^N$ as in Equation 9
7:     Compute the partial coverage reward $f_t(\theta_t) = -\sum_{i=1}^N r_i^{(t-1)} d_i(\theta_t)$
8:     **for** $i = 1, \ldots, N$ **do**
9:         $c_i^{(t)} \leftarrow \min\{c_i^{(t-1)}, d_i(\theta_t)\}$
10:     **end for**
11:     **for** $i = 1, \ldots, N$ **do**
12:         $r_i^{(t)} \leftarrow c_i^{(t)} / \sum_{\ell=1}^N c_\ell^{(t)}$
13:     **end for**
14:     Update the BO history as $\mathcal{H}^{(t)} = \mathcal{H}^{(t-1)} \cup \{(\theta_t, f_t(\theta_t))\}$
15: **end for**
16: **return** Parameters and generated dataset point clouds $\{(\theta_t, \{\mathbf{x}_j(\theta_t)\}_{j=1}^m)\}_{t=1}^T$

---

**Algorithm 3** ROBUST AND DOMAIN-ADAPTED TABULAR FOUNDATION MODELS (RAD-TFM)

---

**Require:** Initial TFM weights $\mathbf{W}$, baseline estimators $\{f_1, \ldots, f_e\}$, $n_e, n_{iter}, n_t, n_d, c \in (0,1)$, reference distribution $P$ from Algorithm 1, parameter space $\Theta$
1: $\{(\theta_i, \delta_i)\}_{i=1}^{n_{trials}} \leftarrow \text{ParameterSearch}(\Theta, \mathbf{W}, \{f_1, \ldots, f_e\}, n_t, n_d)$
2: $\rho \leftarrow c \cdot \rho_{\max}$
3: $\eta \leftarrow \text{RootFinding}(\rho, P, \delta)$     // Find correct temperature for distribution
4: Construct distribution $Q_i \propto p_i \exp(\eta \cdot \delta_i)$
5: **for** $e \in [n_e]$ **do**
6:     **for** $t \in [n_{iter}]$ **do**
7:         $\theta_1, \ldots, \theta_{n_b} \sim Q$     // Sample parameters
8:         $\phi_1, \ldots, \phi_{n_b} \sim p(\Phi; \theta_1), \ldots, p(\Phi; \theta_{n_b})$
9:         $\{(x^i, y^i)\}_{i=1}^{n_b} \sim p(\phi_1), \ldots, p(\phi_{n_b})$     // Sample synthetic datasets
10:         $\{x_{train}^i, x_{test}^i, y_{train}^i, y_{test}^i\}_{i=1}^{n_b} \leftarrow \text{TrainTestSplit}(\{(x^i, y^i)\}_{i=1}^{n_b})$
11:         $l(g_\mathbf{W}) \leftarrow \mathcal{L}_{PFN}(g_\mathbf{W}, \{x_{train}^i, x_{test}^i, y_{train}^i, y_{test}^i\}_{i=1}^{n_b})$
12:         $\mathbf{W} \leftarrow \text{update}(\mathbf{W}, l(g_\mathbf{W}))$     // Gradient step
13:     **end for**
14:     $\{(\theta_i, \delta_i)\}_{i=1}^{n_{trials}} \leftarrow \text{ParameterSearch}(\mathbf{W}, \{f_1, \ldots, f_e\}, n_t, n_d)$
15:     $\eta \leftarrow \text{RootFinding}(H_{min}, \delta)$
16:     Construct distribution $Q_i \propto p_i \exp(\eta \cdot \delta_i)$
17: **end for**
18: **Return** Robust TFM model $g_\mathbf{W}$

---

optimality gaps, as in Equation 6, and a minimization stage, where we sample generators from an adversarial distribution to continue training the TFM. In Algorithm 4, we reference an abstract function SuggestParameters which represents any black-box optimization algorithm. In this case, we use the Optuna python package (Akiba et al., 2019) and the Tree-structed Parzen Estimator (Watanabe, 2023) algorithm, which belongs to the class of Bayesian Optimization algorithms (Frazier, 2018). At each iteration, this algorithm uses the information gained from the previous trials (the estimated optimality gaps) to propose a new parameterization, balancing exploration and exploitation to efficiently find parameters corresponding to large optimality gaps. To begin the minimization stage, we first construct the distribution $Q \in \Delta_\mathcal{P}$. Rather than constructing $\rho$ as a parameter itself, we introduce a scalar parameter $c \in (0,1)$ and define $\rho = c \cdot \rho_{\max}$, such that $\rho$ is a given as a fraction of the maximum possible KL-divergence of $Q$ given the reference distribution $P$. With $\rho_{\max}$ specified, we can find $\eta$ such

---

**Algorithm 4** ParameterSearch

---

**Require:** TFM weights $\mathbf{W}$, baseline estimators $\{f_1, \ldots, f_e\}$, num. trials $n_t$, num. datasets $n_d$

1:   $D_\theta = \{\}$
2:   **for** $i \in [n_t]$ **do**
3:     $\theta_i \leftarrow \text{SuggestParameters}(D_\theta)$           // Black-box optimizer proposes new parameter
4:     **for** $j \in [n_d]$ **do**
5:       $\phi_j \sim p(\Phi; \theta_i)$
6:       $x^j_{train}, x^j_{test}, y^j_{train}, y^j_{test} \sim p(\phi_j)$
7:       $l_j(g_\mathbf{W}) \leftarrow \mathcal{L}_{PFN}(g_\mathbf{W}, x^j_{train}, y^j_{train}, x^j_{test}, y^j_{test})$
8:       $l_b \leftarrow \infty$
9:       **for** $f_k \in \{f_1, \ldots, f_e\}$ **do**
10:         $l_j(f_k) \leftarrow \mathcal{L}_{PFN}(f_k, x^j_{train}, y^j_{train}, x^j_{test}, y^j_{test})$
11:         $l_b \leftarrow \min(l_b, l_j(f_k))$
12:       **end for**
13:       $\delta^j_i \leftarrow l_j(g_\mathbf{W}) - l_b$            // Estimate optimality gap
14:     **end for**
15:     $\delta_i \leftarrow \frac{1}{n_d} \sum_{j=1}^{n_d} \delta^j_i$
16:     $D_\theta \leftarrow D_\theta \cup \{(\theta_i, \delta_i)\}$
17: **end for**
18: **Return** Parameters and estimated optimality gaps $D_\theta$

---

that $D_{KL}(Q(\eta)\|P) = \rho_{\max}$ using one-dimensional root finding (e.g. bisection). Then we generate each dataset in a training batch by sampling a parameter $\theta_i \sim Q$ and generator $\phi \sim p(\Phi; \theta_i)$. We train the model for a fixed number of iterations and then return to the maximization stage. In our implementation, after five epochs, we incorporate the original TabPFN model as another baseline model. This can potentially address regions of the parameter space where the model has reduced its performance by focusing on improving in other regions.

The RAD-TFM algorithm is highly parallelizable, both in the dataset generation steps and the parameter search. For a fixed parameterization $\theta$, each dataset generator $\phi \sim p(\Phi; \theta)$ is sampled independently from the same distribution, such that all datasets generated for a given batch can be created concurrently. The ParameterSearch algorithm can be completely parallelized within each trial. In our implementation, for each trial, we generate a batch of datasets and then fit and evaluate each model and dataset pair concurrently. For example, in our experiments, we set $n_d = 20$ and used $e = 7$ baseline models, resulting in $n_{ds} * e = 140$ separate processes distributed across all available cores. In practice, the domain-adaptation stage is lightweight, while the overall training cost scales approximately linearly with the number of max-min iterations; we provide benchmark-specific compute details in Appendix E.

## C. Theoretical Properties

In this section, we provide the proof that the optimal distribution for the maximizer in equation 6 is a softmax distribution such that $q_i^* \propto p_i \exp(\eta \cdot \delta_{\theta_i}(\mathbf{W}))$. Further, we show that in the absence of a reference distribution, if we take $P$ to be uniform, our formulation can be written as an entropy-constrained DRO, where $Q$ is constrained to have a certain minimum entropy.

**Proposition C.1.** *Let $\delta = (\delta_{\theta_1}(\mathbf{W}), \ldots, \delta_{\theta_n}(\mathbf{W})) \in \mathbb{R}^n$ and let $P = (p_1, \ldots, p_n) \in \Delta_n$ satisfy $p_i > 0$ for all $i$. Consider the distributionally robust inner problem*

$$\max_{Q \in \Delta_n} \sum_{i=1}^{n} q_i \delta_{\theta_i}(\mathbf{W}) \quad s.t. \quad D_{\mathrm{KL}}(Q\|P) \le \rho,$$

*Let $\mathcal{A} = \arg\max_{i \in [n]} \delta_{\theta_i}(\mathbf{W})$, and $P(\mathcal{A}) = \sum_{i \in \mathcal{A}} p_i$. Assume that $\delta_{\theta_i}(\mathbf{W})$ is not constant across $i$ and $0 < \rho < -\log P(\mathcal{A})$. Then the following properties hold.*

(i) *The optimizer $Q^*$ is strictly positive, and there exists a unique temperature parameter $\eta > 0$ such that*

$$q_i^* = \frac{p_i \exp\left(\delta_{\theta_i}(\mathbf{W})/\eta\right)}{\sum_{j=1}^n p_j \exp\left(\delta_{\theta_j}(\mathbf{W})/\eta\right)}, \qquad i = 1, \ldots, n,$$

*where $\eta$ is determined implicitly by $D_{\mathrm{KL}}(Q^*\|P) = \rho$.*

(ii) *If $P$ is the uniform distribution on $[n]$, then the KL constraint $D_{\mathrm{KL}}(Q\|P) \leq \rho$ is equivalent to the entropy constraint $H(Q) \geq \log n - \rho$. In this case, the optimizer in part (i) reduces to the softmax distribution*

$$q_i^* = \frac{\exp\left(\delta_{\theta_i}(\mathbf{W})/\eta\right)}{\sum_{j=1}^n \exp\left(\delta_{\theta_j}(\mathbf{W})/\eta\right)}, \qquad i = 1, \ldots, n,$$

*with $\eta > 0$ chosen so that $H(Q^*) = \log n - \rho$.*

*Proof.* **Part (i)**. For notational simplicity, write $\delta_i = \delta_{\theta_i}(\mathbf{W})$. The problem can be rewritten as

$$\min_{Q \in \Delta_n} -\sum_{i=1}^n q_i \delta_i \quad \text{s.t.} \quad D_{\mathrm{KL}}(Q\|P) \leq \rho.$$

Since $P \in \Delta_n$ satisfies $p_i > 0$ for all $i$ and

$$D_{\mathrm{KL}}(P\|P) = 0 < \rho,$$

Slater's condition holds. Therefore, the KKT conditions are necessary and sufficient.

The Lagrangian is

$$\mathcal{L}(Q, \eta, \mu, \gamma) = -\sum_{i=1}^n q_i \delta_i + \eta\left(\sum_{i=1}^n q_i \log \frac{q_i}{p_i} - \rho\right) + \mu\left(\sum_{i=1}^n q_i - 1\right) - \sum_{i=1}^n \gamma_i q_i,$$

where $\eta \geq 0$, $\mu \in \mathbb{R}$, and $\gamma_i \geq 0$.

We first show that the KL constraint must be active. If the KL constraint were inactive, then $\eta = 0$. In that case, the solution would have to maximize the linear objective over the simplex without an active KL constraint, and hence would place all mass on the set

$$\mathcal{A} = \arg\max_{i \in [n]} \delta_i.$$

Among all distributions supported on $\mathcal{A}$, the one with smallest KL divergence from $P$ is the conditional distribution

$$q_i = \frac{p_i}{P(\mathcal{A})} \mathbf{1}\{i \in \mathcal{A}\},$$

whose KL divergence is

$$D_{\mathrm{KL}}(Q\|P) = \sum_{i \in \mathcal{A}} \frac{p_i}{P(\mathcal{A})} \log \frac{p_i/P(\mathcal{A})}{p_i} = -\log P(\mathcal{A}).$$

Thus, if $\rho < -\log P(\mathcal{A})$, no distribution supported entirely on $\mathcal{A}$ is feasible. Therefore the unconstrained maximizer is infeasible, so the KL constraint must be active and $\eta > 0$.

For an interior optimum, complementary slackness gives $\gamma_i = 0$. The stationarity condition is then

$$-\delta_i + \eta\left(1 + \log \frac{q_i}{p_i}\right) + \mu = 0.$$

Rearranging,

$$\log \frac{q_i}{p_i} = \frac{\delta_i - \mu}{\eta} - 1.$$

Exponentiating gives

$$q_i = p_i \exp\left(\frac{\delta_i}{\eta}\right) \exp\left(-\frac{\mu}{\eta} - 1\right).$$

Enforcing $\sum_i q_i = 1$ yields

$$q_i(\eta) = \frac{p_i \exp(\delta_i/\eta)}{Z(\eta)}, \qquad Z(\eta) = \sum_{j=1}^{n} p_j \exp(\delta_j/\eta).$$

Since $p_i > 0$ and $\eta > 0$, this distribution is strictly positive.

It remains to show that there is a unique $\eta > 0$ satisfying $D_{\mathrm{KL}}(Q(\eta)\|P) = \rho$. Let $E(\eta) = \sum_{i=1}^{n} q_i(\eta)\delta_i$. Then, For the distribution $Q(\eta)$,

$$\log \frac{q_i(\eta)}{p_i} = \frac{\delta_i}{\eta} - \log Z(\eta),$$

so

$$D_{\mathrm{KL}}(Q(\eta)\|P) = \sum_{i=1}^{n} q_i(\eta)\left(\frac{\delta_i}{\eta} - \log Z(\eta)\right) = \frac{E(\eta)}{\eta} - \log Z(\eta).$$

Next,

$$\frac{d}{d\eta} \log Z(\eta) = \frac{1}{Z(\eta)} \sum_{i=1}^{n} p_i \exp(\delta_i/\eta)\left(-\frac{\delta_i}{\eta^2}\right) = -\frac{E(\eta)}{\eta^2}.$$

And,

$$\frac{dE}{d\eta} = -\frac{\mathrm{Var}_{Q(\eta)}(\delta)}{\eta^2}.$$

Therefore,

$$\begin{aligned}
\frac{d}{d\eta} D_{\mathrm{KL}}(Q(\eta)\|P) &= \frac{1}{\eta}\frac{dE}{d\eta} - \frac{E(\eta)}{\eta^2} - \frac{d}{d\eta}\log Z(\eta) \\
&= \frac{1}{\eta}\frac{dE}{d\eta} - \frac{E(\eta)}{\eta^2} + \frac{E(\eta)}{\eta^2} \\
&= -\frac{\mathrm{Var}_{Q(\eta)}(\delta)}{\eta^3}.
\end{aligned}$$

Since $\delta_i$ is not constant across $i$ and $Q(\eta)$ is strictly positive, we have

$$\mathrm{Var}_{Q(\eta)}(\delta) > 0$$

for every $\eta > 0$. We can then conclude that $\eta \mapsto D_{\mathrm{KL}}(Q(\eta)\|P)$ is strictly decreasing on $(0, \infty)$.

Further, we have $\lim_{\eta \to \infty} Q(\eta) = P$, so $\lim_{\eta \to \infty} D_{\mathrm{KL}}(Q(\eta)\|P) = 0$. As $\eta \downarrow 0$, $Q(\eta)$ converges to $P$ conditioned on the set of maximizers $\mathcal{A}$, namely

$$q_i(0) = \frac{p_i}{P(\mathcal{A})}\mathbf{1}\{i \in \mathcal{A}\},$$

and therefore

$$\lim_{\eta \downarrow 0} D_{\mathrm{KL}}(Q(\eta)\|P) = -\log P(\mathcal{A}).$$

Thus, for every $0 < \rho < -\log P(\mathcal{A})$ there exists a unique $\eta > 0$ such that $D_{\mathrm{KL}}(Q(\eta)\|P) = \rho$. By the KKT conditions, this $Q(\eta)$ is the global optimizer.

**Part (ii).** Suppose $P$ is uniform, so $p_i = 1/n$ for all $i$. Then

$$
\begin{aligned}
D_{\mathrm{KL}}(Q\|P) &= \sum_{i=1}^{n} q_i \log \frac{q_i}{1/n} \\
&= \sum_{i=1}^{n} q_i \log q_i + \sum_{i=1}^{n} q_i \log n \\
&= \sum_{i=1}^{n} q_i \log q_i + \log n \\
&= \log n - H(Q).
\end{aligned}
$$

Therefore, $D_{\mathrm{KL}}(Q\|P) \leq \rho$ is equivalent to $\log n - H(Q) \leq \rho$, or $H(Q) \geq \log n - \rho$. Finally, substituting $p_i = 1/n$ into the solution from part (i) gives

$$
q_i^* = \frac{(1/n)\exp(\delta_i/\eta)}{\sum_{j=1}^{n}(1/n)\exp(\delta_j/\eta)} = \frac{\exp(\delta_i/\eta)}{\sum_{j=1}^{n}\exp(\delta_j/\eta)},
$$

with $\eta > 0$ chosen so that $D_{\mathrm{KL}}(Q^*\|P) = \rho$. Using $D_{\mathrm{KL}}(Q^*\|P) = \log n - H(Q^*)$, this is equivalently $H(Q^*) = \log n - \rho$. This proves the claim. $\qquad\square$

## D. SCM Parameters

We summarize the MLP-based SCM parameter space used in our experiments in Table 2. In this work, we instantiate SCMs as randomized MLPs, so the search space includes both architectural parameters and dataset-level properties. The MLP-specific parameters are the hidden width $\mu_h$, number of layers $\mu_l$, input width $\mu_{x_{\mathrm{in}}}$, activation function $\mu_a$, shuffled categorical ratio $\mu_{p_{\mathrm{shuffle}}}$, node noise scale $\mu_{\mathrm{noise}}$, and layer sparsity $\mu_{\mathrm{sparsity}}$. These are combined with shared dataset parameters controlling the number of observed features $\mu_{z_x}$, number of classes $\mu_{n_c}$, mean number of categories $\mu_{\mathrm{mean\_cat}}$, categorical feature ratio $\mu_{p_{\mathrm{cat}}}$, missing-value ratio $\mu_{p_{\mathrm{missing}}}$, missingness mechanism $\mu_{\mathrm{missingness\_type}}$, class-balance concentration $\mu_{\mathrm{class\_balance}}$, weight scaling $\mu_{\mathrm{weight\_scaling}}$, and input distribution $\mu_{\mathrm{dist}}$. Integer-valued parameters are sampled from truncated normal distributions over the ranges shown in Table 2 and then rounded to the nearest integer, while continuous parameters are sampled from truncated normals on their admissible intervals. Categorical parameters are sampled with an $\epsilon$-greedy scheme: with probability $1 - \epsilon$ we select the reference value and with probability $\epsilon$ we sample uniformly from the remaining choices. We also implemented Tree-based SCMs, as in related literature, but found that they did not make a significant contribution to our RAD-TFM pipeline, so we exclude them here for brevity (Qu et al., 2025).

## E. Additional Experiment Details

**Full experiment results.** In Table 3, we report the full results from all six benchmarks. The first three results reported are for the TabArena ($N = 21$), TabPertNet ($N = 70$), and PMLB ($N = 91$) benchmarks. Across our three primary metrics, we observe that RAD-TFM improves performance over all baselines, including the original TabPFN-V2 and the finetuned versions. To measure the significance of our results, we first perform the Friedman test for repeated samples on the median normalized AUC scores (Demšar, 2006; Benavoli et al., 2016). For the TabPertNet datasets, the Friedman test yielded $p = 3.5 \times 10^{-15}$, for TabArena, the test yielded $p = 4.3 \times 10^{-13}$, and for PMLB, the test yielded $p = 5.2 \times 10^{-43}$. Having established significance for the benchmark performance varying by model class, we performed the Wilcoxon signed-rank test pair-wise, comparing the RTFM version of TabPFN to the other models (Conover, 1999). We report the full results from these significance tests in Appendix E. We add an asterisk in this table to indicate results for which the all Wilcoxon signed-rank tests between RAD-TFM and the alternative models achieves $p < 0.05$. For the main benchmarks reported in Table 3, we see that all results are significant for RAD-TFM for TabPertNet and TabArena. Further, for PMLB, the largest p-value obtained was $p = 0.06$, between RAD-TFM and TabPFN-V2 (FT), with $p < 0.05$ for all other pair-wise tests. Our results are therefore almost strictly statistically significant, with RAD-TFM achieving the best performance for each metric across a variety of challenging tabular benchmarks.

We also report results from experiments applying RAD-TFM to domain-specific benchmarks in Engineering ($N = 20$), Genomics ($N = 11$), and Healthcare ($N = 31$), derived from OpenML. To construct these benchmarks, we aggregate all

*Table 2.* MLP-based SCM parameter space and sampling distributions. Numeric parameters are sampled from truncated normal distributions centered at the corresponding mean hyperparameter. Categorical parameters are sampled with an $\epsilon$-greedy rule with $\epsilon = 0.3 \cdot \sigma$.

| Parameter | Type | Range | Distribution |
|---|---|---|---|
| Mean hidden size $\mu_h$ | integer | [1, 256] | $\text{TruncNorm}(\mu_h, 10 \cdot \sigma)$ |
| Mean number of layers $\mu_l$ | integer | [1, 12] | $\text{TruncNorm}(\mu_l, \sigma)$ |
| Mean input width $\mu_{x_{\text{in}}}$ | integer | [1, 15] | $\text{TruncNorm}(\mu_{x_{\text{in}}}, 2 \cdot \sigma)$ |
| Activation function $\mu_a$ | categorical | {relu, elu, tanh, identity} | $\epsilon$-greedy |
| Mean number of features $\mu_{z_x}$ | integer | [2, 200] | $\text{TruncNorm}(\mu_{z_x}, s_{z_x} \cdot \sigma)$ |
| Mean number of classes $\mu_{n_c}$ | integer | [2, 10] | $\text{TruncNorm}(\mu_{n_c}, \sigma)$ |
| Mean number of categories $\mu_{\text{mean\_cat}}$ | integer | [2, 120] | $\text{TruncNorm}(\mu_{\text{mean\_cat}}, 2 \cdot \sigma)$ |
| Categorical feature ratio $\mu_{p_{\text{cat}}}$ | continuous | [0, 1] | $\text{TruncNorm}(\mu_{p_{\text{cat}}}, 0.1 \cdot \sigma)$ |
| Shuffled categorical ratio $\mu_{p_{\text{shuffle}}}$ | continuous | [0, 1] | $\text{TruncNorm}(\mu_{p_{\text{shuffle}}}, 0.1 \cdot \sigma)$ |
| Missing-value ratio $\mu_{p_{\text{missing}}}$ | continuous | [0, 1] | $\text{TruncNorm}(\mu_{p_{\text{missing}}}, 0.1 \cdot \sigma)$ |
| Missingness type $\mu_{\text{missingness\_type}}$ | categorical | {mcar, featurewise, self_masking} | $\epsilon$-greedy |
| Class-balance concentration $\mu_{\text{class\_balance}}$ | continuous | [0.1, 10] | $\text{TruncNorm}(\mu_{\text{class\_balance}}, 1.5 \cdot \sigma)$ |
| Node noise scale $\mu_{\text{noise}}$ | continuous | [0, 1] | $\text{TruncNorm}(\mu_{\text{noise}}, 0.1 \cdot \sigma)$ |
| Layer sparsity $\mu_{\text{sparsity}}$ | continuous | [0, 0.9] | $\text{TruncNorm}(\mu_{\text{sparsity}}, 0.1 \cdot \sigma)$ |
| Weight scaling $\mu_{\text{weight\_scaling}}$ | categorical | {None, Small, Large} | $\epsilon$-greedy |
| Input distribution $\mu_{\text{dist}}$ | categorical | {exponential, uniform, normal} | $\epsilon$-greedy |

binary and multi-task classification datasets on OpenML matching a given tag (e.g. Engineering), then filter to datasets with at least $n = 200$ samples, no more than $n = 10k$ samples, and at most $d = 500$ features. These benchmarks can be highly domain-specific, and differ significantly from common benchmarks such as TabArena. Similar to the other benchmarks, applying RAD-TFM improves the overall performance of TabPFN-V2 across domains, improving mean rank, normalized AUC, and outright wins. Additionally, for these more specific domains, finetuning TabPFN does not exhibit significant benefits. Further, TabPFN (RAD-TFM) strictly dominates the original TabPFN and TabPFN (FT) in rank-1 wins, with the other two receiving zero rank-1 wins across all three domains. The Friedman test returned $p = 1.3 \times 10^{-11}$, $p = 1.3 \times 10^{-5}$, and $p = 4.1 \times 10^{-14}$ for Healthcare, Genomics, and Engineering respectively. Although our approach moves TabPFN's performance in the right direction for these domains, the Wilcoxon signed-rank tests indicate that RAD-TFM only has statistically significant ($p < 0.05$) improvement in normalized AUC for the Engineering benchmark. This is due in part to the small Genomics benchmark ($N = 11$). The Healthcare benchmark is particularly challenging, indicating that it may be necessary to expand either the SCM parameter space or the meta-features used to represent the datasets to enable RAD-TFM to properly target this domain. Overall, these results indicate the potential for RAD-TFM to go beyond finetuning TFMs for domain-specific applications, noticeably improving model performance across three diverse domains.

**RAD-TFM outperforms Fine-Tuning.** To further investigate the impact of finetuning versus our RAD-TFM approach, in Table 4, we compare these two methods against the original TabPFN V2 model in isolation. Across the three benchmarks, we report the number of outright wins for each method and the change in AUC. Additionally, we report a metric we refer to as the *number of leaps*, defined as the number of datasets for which each method causes the resulting TabPFN model to surpass at least one other baseline model, besides the original TabPFN-V2 model itself. This metric captures how often the method not only improves upon TabPFN-V2 incrementally, but improves it enough to surpass alternative model classes. This metric is key to our analysis, given that a major motivation for this work is the observation that, although TFMs are competitive with traditional methods, they do not consistently outperform them. In Table 4, we can see that applying RAD-TFM not only significantly improves the average change in AUC over finetuning alone, but also dominates in the number of wins and, importantly, the number of leaps. In the most extreme case, TabPertNet, finetuning has almost no effect on performance, while applying RAD-TFM leads to 18 leaps, such that TabPFN would now outperform at least one alternative method on 25.7% of datasets in the benchmark. Additionally, we report the standard deviation of normalized AUC scores for each model, which indicates the stability of its performance across datasets. We observe that the robust training process of RAD-TFM reduces the variance in performance across benchmarks, leading to more stable and reliable performance.

**Different Domains Require Different SCM Parameters.** In Table 5, we show a subset of the parameters used to configure the SCM generators, which determine the shape, geometry, and learnability of the generated datasets. For each benchmark, we show the parameters corresponding to the setting with the highest weight, as in Equation 6, the optimal response of the adversary when constrained to generate data that is similar to the real-world reference data. In the table, SCM size refers the

dimensions of the MLP generating the data, which we split into five categorical values, balance refers to the class imbalance (higher corresponds to greater class balance), and noise refers to the scale of the noise added to the output of each neuron. We can clearly observe the variety in parameterizations across benchmarks, some of which align with what we know about the real-world datasets from that domain. For example, for the Genomics benchmark, most datasets have $d = 500$ numeric features, and the domain-adaptation algorithm has correctly constrained the parameter space to datasets with many features (we limit the number of generated features to $d = 200$ per dataset). While some parameters, such as the number of features, may be easier to interpret and adjust manually, for others, such as network sparsity or the activation function, it would be difficult to interpret or understand how changing these values would alter the generated datasets. We can see however, both in Table 5 and Figure 2, that our domain-adaptation algorithm is not only able to adjust these parameters automatically to align with and cover the real-world datasets in the meta-feature space, but that it is necessary to do so, as their values vary significantly between domains.

In Tables 6-13 we provide the full experiment results from the TabPertNet (Ye et al., 2024), TabArena (Erickson et al., 2025), PMLB (Olson et al., 2017), and domain-specific OpenML (Bischl et al., 2021) tabular benchmarks. For all benchmarks, we evaluate on classification datasets only, and limit our evaluation to datasets with at most $n = 10k$ samples and $p = 500$ features. We use the training data from each dataset during the domain-adaptation process to approximate the reference distribution for the given domain. We report our test results on $N = 70$ datasets from TabPertNet, $N = 91$ datasets from PMLB, $N = 21$ datasets from TabArena, $N = 31$ dataset for Healthcare, $N = 11$ datasets for Genomics, and $N = 20$ datasets for Engineering. The original genomics datasets have $O(10k)$ features, so we first preprocess these datasets by randomly selecting a subset of $d = 500$ features and down-sampling. For TabArena, we use the same folds as defined in the original work and corresponding leaderboard. For the OpenML datasets, we use the same train/test splits as provided from the database. For TabPertNet, we use the same train/test split used in the original paper. We report ROC-AUC for binary classification tasks and AUC one-vs-one (OVO) for multi-class tasks.

**Compute Resources.** All experiments were run on a single 80GB A100 GPU together with up to 140 CPU cores during the robust parameter-search stage. Constructing the domain-adapted reference distribution for a benchmark required approximately 10 minutes of GPU time, while each max-min iteration of the RAD-TFM algorithm required approximately 1 GPU-hour; we used 5 epochs for Genomics, 10 epochs for Engineering and Healthcare, and 30 epochs for TabPertNet, TabArena, and PMLB. The CPU-intensive parameter-search stage used roughly 140 cores for about 6 hours per full training run.

**Licensing for existing assets.** Our experiments use existing open-source benchmarks, datasets, models, and software packages. We cite the original sources for each benchmark and model in the main paper. TabArena is used under its publicly released benchmark resources; its code repository is licensed under Apache License 2.0, and the associated dataset curation repository is licensed under the BSD 3-Clause License. The TabArena authors further state that datasets whose licenses do not allow sharing or use in an academic benchmark are excluded from the benchmark. PMLB is publicly available under the MIT License. OpenML data and metadata are made available under OpenML's terms of use; OpenML states that empirical data and metadata are available under a CC-BY license, while individual dataset metadata include dataset-specific license fields. For TabPertNet, we use the publicly released dataset split from the original benchmark; the TP-BERTa (Yan et al., 2024) repository distributing the TabPertNet/OpenTabs subset states that the authors obtained permission to distribute the used subset. We use TabPFN-V2 weights under the Apache 2.0 license. Optuna is licensed under the MIT License. We use all assets only for academic research and benchmarking, and we do not redistribute modified versions of these assets.

*Table 3.* Full experiment results across six tabular dataset benchmarks. For consistency, we report normalized AUC, with values scaled to [0,1] for each dataset (McElfresh et al., 2024; Hollmann et al., 2025). For rank-1 wins, we do not count ties, only outright wins.

| Benchmark | Metric | LR | MLP | RF | CatBoost | XGB | TabPFN$_{n.e.}$ | TabPFN$_{n.e.}$ (FT) | TabPFN$_{n.e.}$ (RAD-TFM) |
|---|---|---|---|---|---|---|---|---|---|
| **TabPertNet**(Ye et al., 2024) | Mean rank AUC | 5.4 | 6.0 | 4.6 | 4.2 | 5.2 | 3.7 | 3.9 | **3.1** |
| | Mean Norm. AUC | 0.4684 | 0.3627 | 0.6646 | 0.7404 | 0.5743 | 0.7306 | 0.7299 | **0.8486**[*] |
| | Rank-1 Wins | 6 | 5 | 7 | 6 | 7 | 9 | 2 | **13** |
| **TabArena**(Erickson et al., 2025) | Mean rank AUC | 5.7 | 7.2 | 5.5 | 4.0 | 5.2 | 3.6 | 2.7 | **2.1** |
| | Mean Norm. AUC | 0.5070 | 0.2513 | 0.6870 | 0.8214 | 0.7293 | 0.8829 | 0.9215 | **0.9421**[*] |
| | Rank-1 Wins | 2 | 0 | 0 | 1 | 3 | 1 | 4 | **9** |
| **PMLB**(Olson et al., 2017) | Mean rank AUC | 5.8 | 7.0 | 4.5 | 4.3 | 4.7 | 3.3 | 3.3 | **3.1** |
| | Mean Norm. AUC | 0.4497 | 0.2782 | 0.7580 | 0.7798 | 0.7036 | 0.8889 | 0.8894 | **0.9070** |
| | Rank-1 Wins | 8 | 2 | 10 | 8 | 7 | 9 | 4 | **16** |
| **Engineering** | Mean rank AUC | 6.9 | 7.3 | 4.3 | 3.4 | 4.9 | 3.3 | 3.4 | **2.6** |
| | Mean Norm. AUC | 0.2944 | 0.3951 | 0.8405 | 0.8952 | 0.7805 | 0.9004 | 0.8998 | **0.9313**[*] |
| | Rank-1 Wins | 0 | 0 | 1 | 4 | 2 | 0 | 0 | **7** |
| **Genomics** | Mean rank AUC | 7.2 | 5.9 | 4.8 | 3.4 | 4.9 | 3.5 | 3.3 | **2.7** |
| | Mean Norm. AUC | 0.1481 | 0.4309 | 0.7439 | 0.8611 | 0.6979 | 0.8801 | 0.8805 | **0.9088** |
| | Rank-1 Wins | 0 | 0 | 1 | 0 | 0 | 0 | 0 | **2** |
| **Healthcare** | Mean rank AUC | 5.3 | 6.9 | 4.5 | 3.5 | 5.0 | 3.8 | 3.8 | **3.2** |
| | Mean Norm. AUC | 0.5831 | 0.2050 | 0.7245 | 0.8446 | 0.6719 | 0.8696 | 0.8699 | **0.8974** |
| | Rank-1 Wins | 2 | 0 | 2 | 2 | 4 | 1 | 0 | **7** |

*Table 4.* Experiment results across three tabular dataset benchmarks, comparing the impact of fine-tuning (FT) compared to our RAD-TFM approach on the TabPFN-V2 Classifier. The #Leaps indicates the number of times the method improves over at least one of the baseline models.

| Benchmark | Metric | TabPFN$_{n.e.}$ | TabPFN$_{n.e.}$ (FT) | TabPFN$_{n.e.}$ (RAD-TFM) |
|---|---|---|---|---|
| **TabPertNet**(Ye et al., 2024) | #Wins | 11 | 5 | **37** |
| | AUC Change (%) | - | $-0.001$ | **2.061** |
| | #Leaps | - | 1 | **18** |
| | Norm. AUC (Std ↓) | 0.345 | 0.343 | **0.215** |
| **TabArena**(Erickson et al., 2025) | #Wins | 1 | 6 | **13** |
| | AUC Change (%) | - | 0.166 | **0.462** |
| | #Leaps | - | 3 | **7** |
| | Norm. AUC (Std ↓) | 0.223 | 0.195 | **0.178** |
| **PMLB**(Olson et al., 2017) | #Wins | 15 | 8 | **37** |
| | AUC Change (%) | - | 0.016 | **0.332** |
| | #Leaps | - | 1 | **8** |
| | Norm. AUC (Std ↓) | 0.210 | 0.209 | **0.188** |

*Table 5.* SCM generator parameters with the highest weight in the first iteration of the RAD-TFM algorithm for various benchmarks. Different benchmarks correspond to significantly different data generation regimes. We report only a subset of the parameters in this table.

| Benchmark | SCM Size | # Feat. | # Class | Act. | Sparsity | $p_{cat}$ | Cat. Dim. | $p_{miss}$ | Balance | Noise |
|---|---|---|---|---|---|---|---|---|---|---|
| **TabPertNet**(Ye et al., 2024) | Medium | 13 | 2 | tanh | 0.9 | 0.0 | - | 0.4 | 3 | 0.0 |
| **TabArena**(Erickson et al., 2025) | Medium | 44 | 2 | tanh | 0.9 | 0.5 | 50 | 0.2 | 3 | 0.0 |
| **PMLB**(Olson et al., 2017) | Small | 12 | 10 | tanh | 0.9 | 0.0 | - | 0.2 | 3 | 0.0 |
| **Engineering** | Large | 65 | 2 | elu | 0.0 | 0.0 | - | 0.0 | 10 | 0.01 |
| **Genomics** | Large | 200 | 2 | relu | 0.1 | 0.0 | - | 0.4 | 3 | 0.05 |
| **Healthcare** | Small | 5 | 2 | identity | 0.0 | 1.0 | 2 | 0.2 | 10 | 0.5 |

*Table 6.* TabPertNet (Ye et al., 2024) Test Results, Part 1 (AUC ↑).

| Dataset | TabPFN$_{n.e.}$ (RAD-TFM) | TabPFN$_{n.e.}$ (Base) | TabPFN$_{n.e.}$ (Finetuned) | CatBoost | RF | XGB | LR | MLP |
|---|---|---|---|---|---|---|---|---|
| BankNoteAuthenticati | 1.000 | 1.000 | 1.000 | 1.000 | 1.000 | 1.000 | 0.999 | 0.998 |
| Bank_Personal_Loan_M | 0.597 | 0.578 | 0.578 | 0.600 | 0.601 | 0.599 | 0.606 | 0.590 |
| Breast_Cancer | 0.882 | 0.876 | 0.876 | 0.864 | 0.891 | 0.839 | 0.885 | 0.869 |
| Customer_Behaviour | 0.958 | 0.958 | 0.958 | 0.944 | 0.968 | 0.972 | 0.944 | 0.940 |
| Employee Satisfactio | 0.655 | 0.433 | 0.441 | 0.543 | 0.473 | 0.436 | 0.439 | 0.598 |
| NFL | 0.725 | 0.731 | 0.731 | 0.720 | 0.679 | 0.671 | 0.670 | 0.685 |
| TravelInsurancePredi | 0.708 | 0.670 | 0.668 | 0.708 | 0.546 | 0.631 | 0.715 | 0.723 |
| UniversalBank | 0.597 | 0.578 | 0.578 | 0.600 | 0.593 | 0.599 | 0.606 | 0.590 |
| audit_data | 0.997 | 0.998 | 0.998 | 1.000 | 1.000 | 1.000 | 0.992 | 0.981 |
| audit_risk | 0.997 | 0.998 | 0.998 | 1.000 | 1.000 | 1.000 | 0.992 | 0.981 |
| b_depressed | 0.468 | 0.447 | 0.445 | 0.578 | 0.574 | 0.513 | 0.446 | 0.601 |
| bt_dataset_t3 | 1.000 | 0.999 | 0.999 | 0.998 | 0.995 | 0.995 | 0.969 | 0.814 |
| loan | 0.673 | 0.659 | 0.659 | 0.669 | 0.691 | 0.697 | 0.677 | 0.654 |
| mechanical_analysis | 1.000 | 1.000 | 1.000 | 1.000 | 1.000 | 1.000 | 0.949 | 0.940 |
| new_model | 1.000 | 1.000 | 1.000 | 1.000 | 1.000 | 1.000 | 1.000 | 1.000 |
| piracydataset | 0.573 | 0.572 | 0.571 | 0.592 | 0.606 | 0.607 | 0.606 | 0.486 |
| 0284_bank8FM | 0.994 | 0.994 | 0.994 | 0.994 | 0.991 | 0.992 | 0.993 | 0.994 |
| 0292_cpu_small | 0.980 | 0.980 | 0.980 | 0.979 | 0.975 | 0.978 | 0.959 | 0.972 |
| 0312_cpu_act | 0.987 | 0.988 | 0.988 | 0.984 | 0.983 | 0.986 | 0.970 | 0.982 |
| 0345_delta_ailerons | 0.977 | 0.975 | 0.975 | 0.974 | 0.975 | 0.972 | 0.967 | 0.971 |
| 0356_delta_elevators | 0.950 | 0.949 | 0.949 | 0.952 | 0.946 | 0.944 | 0.945 | 0.949 |
| 0400_analcatdata_sup | 1.000 | 1.000 | 1.000 | 1.000 | 1.000 | 1.000 | 0.938 | 0.750 |
| 0419_pm10 | 0.643 | 0.438 | 0.433 | 0.567 | 0.677 | 0.570 | 0.404 | 0.562 |
| 0435_strikes | 1.000 | 1.000 | 1.000 | 0.996 | 0.990 | 1.000 | 0.616 | 0.555 |
| 0437_quake | 0.521 | 0.432 | 0.431 | 0.447 | 0.500 | 0.482 | 0.563 | 0.531 |
| 0445_arsenic-male-bl | 0.744 | 0.600 | 0.600 | 0.533 | 0.500 | 0.778 | 0.411 | 0.711 |
| 0446_arsenic-female- | 0.858 | 0.845 | 0.844 | 0.807 | 0.737 | 0.786 | 0.814 | 0.663 |
| 0472_analcatdata_mar | 0.663 | 0.360 | 0.366 | 0.571 | 0.526 | 0.480 | 0.383 | 0.429 |
| 0509_pollen | 0.512 | 0.500 | 0.502 | 0.503 | 0.498 | 0.530 | 0.456 | 0.528 |
| 0526_colleges_aaup | 1.000 | 1.000 | 1.000 | 1.000 | 1.000 | 0.999 | 0.967 | 0.961 |
| 0541_plasma_retinol | 0.585 | 0.591 | 0.591 | 0.636 | 0.699 | 0.352 | 0.625 | 0.511 |
| 0555_socmob | 0.992 | 0.987 | 0.987 | 0.995 | 0.989 | 0.992 | 0.927 | 0.931 |
| 0885_compas-two-year | 0.709 | 0.709 | 0.709 | 0.693 | 0.441 | 0.570 | 0.703 | 0.684 |
| 0948_Ishwar | 0.996 | 0.998 | 0.998 | 0.990 | 0.983 | 0.988 | 0.975 | 0.979 |
| 1011_cleve | 0.943 | 0.933 | 0.933 | 0.929 | 0.943 | 0.933 | 0.957 | 0.971 |
| 1142_Sick_numeric | 0.995 | 0.991 | 0.991 | 0.995 | 0.983 | 0.978 | 0.917 | 0.578 |

*Table 7.* TabPertNet (Ye et al., 2024) Test Results, Part 2 (AUC ↑).

| Dataset | TabPFN$_{n.e.}$ (RAD-TFM) | TabPFN$_{n.e.}$ (Base) | TabPFN$_{n.e.}$ (Finetuned) | CatBoost | RF | XGB | LR | MLP |
|---|---|---|---|---|---|---|---|---|
| 1201_Gender-Recognit | 1.000 | 1.000 | 1.000 | 0.998 | 0.999 | 0.999 | 0.996 | 0.993 |
| 1222_premier_league_ | 0.695 | 0.699 | 0.699 | 0.577 | 0.634 | 0.618 | 0.693 | 0.661 |
| 1408_national-longit | 1.000 | 1.000 | 1.000 | 1.000 | 1.000 | 1.000 | 0.992 | 0.990 |
| 1413_shill-bidding | 1.000 | 1.000 | 1.000 | 1.000 | 1.000 | 1.000 | 0.996 | 0.997 |
| 1458_kdd_ipums_la_97 | 0.938 | 0.937 | 0.937 | 0.939 | 0.932 | 0.923 | 0.890 | 0.909 |
| 1461_heart-failure | 0.861 | 0.861 | 0.861 | 0.822 | 0.844 | 0.856 | 0.944 | 0.794 |
| 1512_eye_movements | 0.714 | 0.746 | 0.745 | 0.690 | 0.698 | 0.707 | 0.594 | 0.627 |
| 1564_Mammographic-Ma | 0.817 | 0.815 | 0.815 | 0.808 | 0.701 | 0.658 | 0.807 | 0.691 |
| 1578_kdd_ipums_la_97 | 0.954 | 0.953 | 0.953 | 0.959 | 0.954 | 0.946 | 0.907 | 0.927 |
| 1592_Diabetes-Data-S | 0.873 | 0.871 | 0.871 | 0.876 | 0.896 | 0.871 | 0.839 | 0.772 |
| 1600_VulNoneVul | 0.801 | 0.798 | 0.797 | 0.720 | 0.638 | 0.698 | 0.755 | 0.523 |
| 1635_Is-this-a-good- | 0.810 | 0.805 | 0.804 | 0.820 | 0.779 | 0.752 | 0.747 | 0.750 |
| 1692_Gender-Classifi | 0.995 | 0.995 | 0.995 | 0.993 | 0.992 | 0.989 | 0.993 | 0.993 |
| 1736_combined-wine-d | 1.000 | 1.000 | 1.000 | 1.000 | 1.000 | 1.000 | 0.999 | 0.998 |
| 1742_Loan-Predicatio | 0.785 | 0.743 | 0.743 | 0.766 | 0.769 | 0.737 | 0.742 | 0.752 |
| 1752_Wisconsin-breas | 0.996 | 0.994 | 0.994 | 0.990 | 0.992 | 0.987 | 0.992 | 0.977 |
| 1759_Red_White-wine- | 1.000 | 1.000 | 1.000 | 1.000 | 1.000 | 1.000 | 0.999 | 0.998 |
| 1898_Personal-Loan-M | 0.618 | 0.624 | 0.623 | 0.631 | 0.645 | 0.603 | 0.627 | 0.587 |
| 2304_electricity_see | 0.875 | 0.866 | 0.866 | 0.853 | 0.867 | 0.875 | 0.699 | 0.770 |
| 2305_electricity_see | 0.934 | 0.937 | 0.937 | 0.905 | 0.917 | 0.922 | 0.788 | 0.858 |
| 2306_electricity_see | 0.877 | 0.879 | 0.879 | 0.862 | 0.874 | 0.871 | 0.771 | 0.827 |
| 2308_electricity_see | 0.912 | 0.911 | 0.911 | 0.891 | 0.886 | 0.886 | 0.717 | 0.837 |
| 2389_airlines_seed_0 | 0.679 | 0.683 | 0.683 | 0.640 | 0.618 | 0.637 | 0.683 | 0.666 |
| 2390_airlines_seed_1 | 0.646 | 0.640 | 0.640 | 0.637 | 0.607 | 0.585 | 0.662 | 0.646 |
| 2391_airlines_seed_2 | 0.650 | 0.641 | 0.641 | 0.619 | 0.618 | 0.650 | 0.647 | 0.635 |
| 2392_airlines_seed_3 | 0.607 | 0.623 | 0.623 | 0.607 | 0.600 | 0.564 | 0.612 | 0.617 |
| 2393_airlines_seed_4 | 0.677 | 0.680 | 0.679 | 0.647 | 0.639 | 0.607 | 0.690 | 0.682 |
| 2619_sf-police-incid | 0.465 | 0.450 | 0.450 | 0.530 | 0.595 | 0.410 | 0.495 | 0.445 |
| 2620_sf-police-incid | 0.509 | 0.487 | 0.485 | 0.518 | 0.510 | 0.457 | 0.512 | 0.456 |
| 2621_sf-police-incid | 0.478 | 0.540 | 0.544 | 0.731 | 0.739 | 0.616 | 0.646 | 0.449 |
| 2622_sf-police-incid | 0.522 | 0.458 | 0.458 | 0.482 | 0.442 | 0.430 | 0.449 | 0.467 |
| 2703_compas-two-year | 0.729 | 0.730 | 0.730 | 0.711 | 0.525 | 0.585 | 0.732 | 0.733 |
| trial | 1.000 | 1.000 | 1.000 | 1.000 | 1.000 | 1.000 | 1.000 | 1.000 |
| wines_SPA | 1.000 | 1.000 | 1.000 | 1.000 | 1.000 | 1.000 | 1.000 | 1.000 |

*Table 8.* TabArena (Erickson et al., 2025) Test Results (OVO AUC ↑).

| Dataset | TabPFN$_{n.e.}$ (RAD-TFM) | TabPFN$_{n.e.}$ (Base) | TabPFN$_{n.e.}$ (Finetuned) | CatBoost | RF | XGB | LR | MLP |
|---|---|---|---|---|---|---|---|---|
| Bank_Customer_Churn | 0.876 | 0.875 | 0.874 | 0.878 | 0.845 | 0.845 | 0.745 | 0.848 |
| Fitness_Club | 0.811 | 0.810 | 0.810 | 0.803 | 0.760 | 0.757 | 0.796 | 0.563 |
| Is-this-a-good-custo | 0.731 | 0.707 | 0.712 | 0.729 | 0.655 | 0.670 | 0.712 | 0.585 |
| MIC | 0.643 | 0.625 | 0.634 | 0.684 | 0.678 | 0.652 | 0.722 | 0.635 |
| Marketing_Campaign | 0.908 | 0.897 | 0.894 | 0.907 | 0.894 | 0.909 | 0.837 | 0.647 |
| NATICUSdroid | 0.986 | 0.984 | 0.985 | 0.985 | 0.978 | 0.984 | 0.982 | 0.981 |
| anneal | 0.992 | 1.000 | 1.000 | 0.958 | 0.999 | 0.994 | 0.953 | 0.961 |
| blood-transfusion-se | 0.741 | 0.729 | 0.732 | 0.697 | 0.663 | 0.649 | 0.744 | 0.723 |
| churn | 0.944 | 0.940 | 0.942 | 0.938 | 0.927 | 0.933 | 0.814 | 0.882 |
| coil2000_insurance_p | 0.752 | 0.746 | 0.749 | 0.721 | 0.689 | 0.702 | 0.735 | 0.696 |
| credit-g | 0.780 | 0.781 | 0.782 | 0.774 | 0.771 | 0.772 | 0.773 | 0.704 |
| diabetes | 0.841 | 0.842 | 0.844 | 0.830 | 0.828 | 0.808 | 0.839 | 0.781 |
| hazelnut-spread-cont | 0.982 | 0.982 | 0.983 | 0.959 | 0.950 | 0.968 | 0.584 | 0.931 |
| maternal_health_risk | 0.962 | 0.952 | 0.952 | 0.952 | 0.955 | 0.958 | 0.804 | 0.793 |
| polish_companies_ban | 0.967 | 0.963 | 0.966 | 0.970 | 0.945 | 0.963 | 0.532 | 0.814 |
| qsar-biodeg | 0.930 | 0.925 | 0.930 | 0.925 | 0.915 | 0.920 | 0.921 | 0.919 |
| seismic-bumps | 0.773 | 0.782 | 0.778 | 0.758 | 0.761 | 0.740 | 0.766 | 0.602 |
| splice | 0.998 | 0.997 | 0.998 | 0.996 | 0.997 | 0.995 | 0.992 | 0.988 |
| students_dropout_and | 0.869 | 0.864 | 0.866 | 0.857 | 0.856 | 0.869 | 0.864 | 0.832 |
| taiwanese_bankruptcy | 0.948 | 0.948 | 0.950 | 0.941 | 0.941 | 0.940 | 0.592 | 0.832 |
| website_phishing | 0.973 | 0.973 | 0.973 | 0.976 | 0.970 | 0.972 | 0.926 | 0.869 |

Table 9. PMLB (Olson et al., 2017) Test Results, Part 1 (OVO AUC ↑).

| Dataset | TabPFN$_{n.e.}$ (RAD-TFM) | TabPFN$_{n.e.}$ (Base) | TabPFN$_{n.e.}$ (Finetuned) | CatBoost | RF | XGB | LR | MLP |
|---|---|---|---|---|---|---|---|---|
| 1028_SWD | 0.841 | 0.843 | 0.843 | 0.861 | 0.861 | 0.861 | 0.860 | 0.827 |
| 1029_LEV | 0.902 | 0.901 | 0.902 | 0.917 | 0.911 | 0.916 | 0.885 | 0.849 |
| 1030_ERA | 0.788 | 0.787 | 0.787 | 0.742 | 0.737 | 0.739 | 0.788 | 0.724 |
| 294_satellite_image | 0.995 | 0.995 | 0.995 | 0.989 | 0.988 | 0.991 | 0.957 | 0.985 |
| GAMETES_Epistasis_2_ | 0.739 | 0.738 | 0.737 | 0.684 | 0.601 | 0.662 | 0.530 | 0.464 |
| GAMETES_Epistasis_2_ | 0.876 | 0.871 | 0.872 | 0.851 | 0.711 | 0.807 | 0.498 | 0.666 |
| GAMETES_Epistasis_3_ | 0.480 | 0.482 | 0.482 | 0.619 | 0.563 | 0.617 | 0.539 | 0.465 |
| GAMETES_Heterogeneit | 0.731 | 0.727 | 0.727 | 0.768 | 0.725 | 0.769 | 0.533 | 0.637 |
| GAMETES_Heterogeneit | 0.795 | 0.783 | 0.785 | 0.745 | 0.620 | 0.668 | 0.513 | 0.535 |
| Hill_Valley_with_noi | 0.532 | 0.511 | 0.510 | 0.486 | 0.558 | 0.467 | 0.496 | 0.502 |
| Hill_Valley_without_ | 0.542 | 0.474 | 0.474 | 0.477 | 0.587 | 0.458 | 0.496 | 0.495 |
| agaricus_lepiota | 1.000 | 1.000 | 1.000 | 1.000 | 1.000 | 1.000 | 0.975 | 1.000 |
| allbp | 0.995 | 0.995 | 0.995 | 0.991 | 0.994 | 0.993 | 0.932 | 0.659 |
| allhypo | 0.986 | 0.984 | 0.984 | 0.982 | 0.983 | 0.981 | 0.840 | 0.707 |
| allrep | 0.904 | 0.899 | 0.899 | 0.898 | 0.929 | 0.890 | 0.821 | 0.456 |
| analcatdata_authorsh | 1.000 | 1.000 | 1.000 | 1.000 | 1.000 | 1.000 | 0.999 | 0.999 |
| analcatdata_dmft | 0.621 | 0.605 | 0.614 | 0.533 | 0.517 | 0.548 | 0.597 | 0.449 |
| analcatdata_germangs | 0.839 | 0.835 | 0.837 | 0.660 | 0.595 | 0.660 | 0.698 | 0.573 |
| analcatdata_lawsuit | 1.000 | 1.000 | 1.000 | 1.000 | 1.000 | 1.000 | 1.000 | 0.987 |
| ann_thyroid | 0.998 | 0.999 | 0.999 | 1.000 | 1.000 | 1.000 | 0.789 | 0.922 |
| balance_scale | 0.985 | 0.986 | 0.986 | 0.915 | 0.754 | 0.871 | 0.941 | 0.852 |
| banana | 0.967 | 0.967 | 0.967 | 0.966 | 0.960 | 0.958 | 0.501 | 0.945 |
| breast_cancer | 0.800 | 0.774 | 0.774 | 0.800 | 0.783 | 0.539 | 0.800 | 0.626 |
| breast_cancer_wiscon | 1.000 | 1.000 | 1.000 | 1.000 | 1.000 | 0.997 | 1.000 | 0.929 |
| breast_cancer_wiscon | 0.985 | 0.986 | 0.986 | 0.992 | 0.987 | 0.984 | 0.986 | 0.997 |
| bupa | 0.671 | 0.667 | 0.667 | 0.650 | 0.625 | 0.496 | 0.688 | 0.667 |
| calendarDOW | 0.855 | 0.848 | 0.846 | 0.836 | 0.861 | 0.843 | 0.801 | 0.779 |
| car_evaluation | 0.999 | 0.999 | 0.999 | 1.000 | 0.999 | 1.000 | 0.921 | 0.862 |
| cars | 1.000 | 1.000 | 1.000 | 0.931 | 0.932 | 0.929 | 0.781 | 0.628 |
| chess | 0.999 | 0.999 | 0.999 | 1.000 | 1.000 | 0.999 | 0.996 | 0.998 |
| churn | 0.974 | 0.973 | 0.973 | 0.964 | 0.960 | 0.969 | 0.769 | 0.580 |
| clean1 | 1.000 | 1.000 | 1.000 | 1.000 | 1.000 | 1.000 | 1.000 | 0.918 |
| clean2 | 1.000 | 1.000 | 1.000 | 1.000 | 1.000 | 1.000 | 1.000 | 1.000 |
| coil2000 | 0.769 | 0.763 | 0.763 | 0.742 | 0.656 | 0.741 | 0.723 | 0.601 |
| congressional_voting | 1.000 | 1.000 | 1.000 | 1.000 | 1.000 | 0.981 | 0.998 | 0.976 |
| contraceptive_method | 0.729 | 0.730 | 0.730 | 0.688 | 0.701 | 0.702 | 0.670 | 0.648 |
| credit_approval_aust | 0.959 | 0.958 | 0.958 | 0.960 | 0.966 | 0.944 | 0.951 | 0.918 |
| credit_approval_germ | 0.844 | 0.846 | 0.846 | 0.829 | 0.798 | 0.733 | 0.824 | 0.735 |

Table 10. PMLB (Olson et al., 2017) Test Results, Part 2 (OVO AUC ↑).

| Dataset | TabPFN$_{n.e.}$ (RAD-TFM) | TabPFN$_{n.e.}$ (Base) | TabPFN$_{n.e.}$ (Finetuned) | CatBoost | RF | XGB | LR | MLP |
|---|---|---|---|---|---|---|---|---|
| dis | 0.999 | 1.000 | 1.000 | 0.996 | 0.995 | 0.998 | 0.732 | 0.771 |
| dna | 0.995 | 0.995 | 0.995 | 0.997 | 0.996 | 0.998 | 0.990 | 0.991 |
| ecoli | 0.960 | 0.960 | 0.960 | 0.955 | 0.956 | 0.947 | 0.941 | 0.872 |
| haberman | 0.725 | 0.683 | 0.683 | 0.608 | 0.529 | 0.375 | 0.717 | 0.875 |
| heart_disease_clevel | 0.904 | 0.894 | 0.894 | 0.904 | 0.916 | 0.904 | 0.904 | 0.880 |
| heart_disease_hungar | 0.911 | 0.900 | 0.900 | 0.947 | 0.945 | 0.895 | 0.905 | 0.574 |
| horse_colic_outcome | 0.764 | 0.749 | 0.753 | 0.812 | 0.828 | 0.792 | 0.731 | 0.776 |
| horse_colic_surgery | 0.821 | 0.810 | 0.810 | 0.790 | 0.804 | 0.722 | 0.841 | 0.587 |
| hypothyroid | 0.967 | 0.968 | 0.968 | 0.979 | 0.979 | 0.977 | 0.865 | 0.546 |
| ionosphere | 1.000 | 0.996 | 0.996 | 1.000 | 1.000 | 1.000 | 0.930 | 0.816 |
| irish | 1.000 | 1.000 | 1.000 | 1.000 | 1.000 | 1.000 | 0.869 | 0.902 |
| kr_vs_kp | 1.000 | 1.000 | 1.000 | 1.000 | 1.000 | 1.000 | 0.993 | 0.998 |
| led24 | 0.948 | 0.948 | 0.948 | 0.941 | 0.942 | 0.929 | 0.948 | 0.946 |
| led7 | 0.957 | 0.957 | 0.957 | 0.949 | 0.947 | 0.952 | 0.958 | 0.955 |
| mfeat_factors | 1.000 | 1.000 | 1.000 | 0.999 | 1.000 | 1.000 | 1.000 | 1.000 |
| mfeat_fourier | 0.992 | 0.993 | 0.993 | 0.986 | 0.986 | 0.988 | 0.981 | 0.977 |
| mfeat_karhunen | 1.000 | 1.000 | 1.000 | 0.999 | 0.999 | 0.999 | 1.000 | 0.999 |
| mfeat_morphological | 0.968 | 0.969 | 0.969 | 0.962 | 0.955 | 0.958 | 0.960 | 0.954 |
| mfeat_pixel | 1.000 | 1.000 | 1.000 | 1.000 | 1.000 | 1.000 | 1.000 | 0.999 |
| mfeat_zernike | 0.990 | 0.990 | 0.990 | 0.974 | 0.973 | 0.966 | 0.982 | 0.983 |
| mofn_3_7_10 | 1.000 | 1.000 | 1.000 | 1.000 | 1.000 | 1.000 | 1.000 | 1.000 |
| monk1 | 1.000 | 1.000 | 1.000 | 1.000 | 1.000 | 1.000 | 0.374 | 0.760 |
| monk2 | 1.000 | 1.000 | 1.000 | 1.000 | 0.968 | 1.000 | 0.559 | 0.544 |
| monk3 | 1.000 | 1.000 | 1.000 | 1.000 | 0.998 | 1.000 | 0.871 | 0.871 |
| mushroom | 1.000 | 1.000 | 1.000 | 1.000 | 1.000 | 1.000 | 0.977 | 1.000 |
| optdigits | 1.000 | 1.000 | 1.000 | 0.999 | 1.000 | 0.999 | 0.999 | 0.999 |
| page_blocks | 0.963 | 0.963 | 0.963 | 0.964 | 0.966 | 0.962 | 0.903 | 0.962 |
| parity5+5 | 1.000 | 1.000 | 1.000 | 0.531 | 0.651 | 1.000 | 0.643 | 0.333 |
| pendigits | 1.000 | 1.000 | 1.000 | 1.000 | 1.000 | 1.000 | 0.996 | 1.000 |
| penguins | 1.000 | 1.000 | 1.000 | 1.000 | 1.000 | 1.000 | 1.000 | 0.901 |
| phoneme | 0.951 | 0.954 | 0.954 | 0.938 | 0.953 | 0.946 | 0.835 | 0.887 |
| profb | 0.698 | 0.670 | 0.671 | 0.652 | 0.671 | 0.680 | 0.656 | 0.579 |
| ring | 0.996 | 0.996 | 0.996 | 0.995 | 0.994 | 0.995 | 0.837 | 0.993 |
| saheart | 0.767 | 0.753 | 0.757 | 0.809 | 0.739 | 0.712 | 0.772 | 0.658 |
| satimage | 0.995 | 0.995 | 0.995 | 0.989 | 0.988 | 0.991 | 0.957 | 0.985 |
| schizo | 0.578 | 0.567 | 0.567 | 0.444 | 0.421 | 0.443 | 0.583 | 0.533 |
| segmentation | 1.000 | 1.000 | 1.000 | 1.000 | 1.000 | 1.000 | 0.995 | 0.994 |
| spambase | 0.988 | 0.989 | 0.989 | 0.988 | 0.980 | 0.990 | 0.976 | 0.971 |
| spect | 0.881 | 0.887 | 0.887 | 0.862 | 0.875 | 0.881 | 0.919 | 0.875 |
| spectf | 0.914 | 0.919 | 0.919 | 0.949 | 0.944 | 0.899 | 0.848 | 0.828 |
| splice | 0.997 | 0.997 | 0.997 | 0.998 | 0.995 | 0.998 | 0.944 | 0.956 |
| threeOf9 | 1.000 | 1.000 | 1.000 | 1.000 | 1.000 | 1.000 | 0.961 | 0.969 |
| tic_tac_toe | 0.998 | 0.999 | 0.999 | 0.999 | 0.995 | 1.000 | 0.602 | 0.778 |
| titanic | 0.805 | 0.804 | 0.804 | 0.783 | 0.728 | 0.742 | 0.713 | 0.705 |
| tokyo1 | 0.980 | 0.977 | 0.977 | 0.987 | 0.984 | 0.984 | 0.883 | 0.951 |
| twonorm | 0.998 | 0.998 | 0.998 | 0.998 | 0.997 | 0.997 | 0.998 | 0.998 |
| vehicle | 0.978 | 0.978 | 0.977 | 0.941 | 0.949 | 0.938 | 0.941 | 0.818 |
| waveform_21 | 0.972 | 0.972 | 0.972 | 0.964 | 0.966 | 0.966 | 0.972 | 0.967 |
| waveform_40 | 0.972 | 0.973 | 0.973 | 0.960 | 0.967 | 0.968 | 0.966 | 0.964 |
| wine_quality_red | 0.814 | 0.822 | 0.822 | 0.821 | 0.871 | 0.849 | 0.777 | 0.673 |
| wine_quality_white | 0.860 | 0.876 | 0.875 | 0.783 | 0.799 | 0.809 | 0.639 | 0.635 |
| xd6 | 1.000 | 1.000 | 1.000 | 1.000 | 1.000 | 1.000 | 0.886 | 0.996 |
| yeast | 0.873 | 0.876 | 0.875 | 0.853 | 0.870 | 0.870 | 0.827 | 0.814 |

*Table 11.* Engineering (Bischl et al., 2021) Test Results (OVO AUC ↑).

| Dataset | TabPFN$_{n.e.}$ (RAD-TFM) | TabPFN$_{n.e.}$ (Base) | TabPFN$_{n.e.}$ (Finetuned) | CatBoost | RF | XGB | LR | MLP |
|---|---|---|---|---|---|---|---|---|
| anneal_2 | 1.000 | 1.000 | 1.000 | 1.000 | 1.000 | 1.000 | 0.970 | 0.997 |
| cnae-9_1468 | 0.999 | 0.998 | 0.998 | 0.998 | 0.997 | 0.997 | 0.998 | 0.993 |
| glass_41 | 0.975 | 0.964 | 0.964 | 0.952 | 0.939 | 0.937 | 0.819 | 0.497 |
| kc1_1067 | 0.868 | 0.861 | 0.861 | 0.840 | 0.849 | 0.802 | 0.789 | 0.820 |
| kc2_1063 | 0.868 | 0.864 | 0.864 | 0.892 | 0.868 | 0.835 | 0.712 | 0.831 |
| kc3_1065 | 0.780 | 0.774 | 0.774 | 0.750 | 0.848 | 0.685 | 0.554 | 0.702 |
| mc1_1056 | 0.912 | 0.922 | 0.921 | 0.990 | 0.922 | 0.992 | 0.917 | 0.884 |
| mw1_1071 | 0.842 | 0.868 | 0.868 | 0.719 | 0.711 | 0.588 | 0.833 | 0.640 |
| pc1_1068 | 0.882 | 0.875 | 0.876 | 0.875 | 0.805 | 0.821 | 0.692 | 0.533 |
| pc2_1069 | 0.990 | 0.990 | 0.990 | 0.993 | 0.982 | 0.985 | 0.035 | 0.250 |
| pc3_1050 | 0.847 | 0.825 | 0.825 | 0.785 | 0.809 | 0.731 | 0.640 | 0.719 |
| pc4_1049 | 0.946 | 0.930 | 0.930 | 0.938 | 0.934 | 0.949 | 0.918 | 0.880 |
| seismic-bumps_1500 | 0.993 | 0.993 | 0.993 | 0.966 | 0.966 | 0.980 | 0.993 | 0.748 |
| semeion_1501 | 0.998 | 0.996 | 0.996 | 0.997 | 0.996 | 0.995 | 0.995 | 0.991 |
| sonar_40 | 0.963 | 0.963 | 0.963 | 0.981 | 0.954 | 0.972 | 0.926 | 0.843 |
| steel-plates-fault_1 | 1.000 | 1.000 | 1.000 | 1.000 | 1.000 | 1.000 | 0.698 | 1.000 |
| steel-plates-fault_4 | 0.980 | 0.980 | 0.980 | 0.974 | 0.973 | 0.971 | 0.737 | 0.950 |
| wall-robot-navigatio | 1.000 | 1.000 | 1.000 | 1.000 | 1.000 | 1.000 | 0.908 | 0.986 |
| wall-robot-navigatio | 1.000 | 1.000 | 1.000 | 1.000 | 1.000 | 1.000 | 0.986 | 1.000 |
| wall-robot-navigatio | 1.000 | 1.000 | 1.000 | 1.000 | 1.000 | 1.000 | 0.987 | 0.999 |

*Table 12.* Genomics (Bischl et al., 2021) Test Results (AUC ↑).

| Dataset | TabPFN$_{n.e.}$ (RAD-TFM) | TabPFN$_{n.e.}$ (Base) | TabPFN$_{n.e.}$ (Finetuned) | CatBoost | RF | XGB | LR | MLP |
|---|---|---|---|---|---|---|---|---|
| AP_Breast_Omentum_11 | 0.914 | 0.914 | 0.914 | 0.882 | 0.882 | 0.879 | 0.866 | 0.864 |
| AP_Colon_Omentum_114 | 1.000 | 1.000 | 1.000 | 0.996 | 0.996 | 1.000 | 0.970 | 1.000 |
| AP_Colon_Ovary_1153 | 0.984 | 0.986 | 0.986 | 0.984 | 0.974 | 0.972 | 0.948 | 0.950 |
| AP_Colon_Uterus_1160 | 0.970 | 0.962 | 0.962 | 0.970 | 0.929 | 0.951 | 0.967 | 0.967 |
| AP_Omentum_Kidney_11 | 0.990 | 0.990 | 0.990 | 0.995 | 0.990 | 0.995 | 0.971 | 0.971 |
| AP_Omentum_Lung_1132 | 0.990 | 0.990 | 0.990 | 0.981 | 0.990 | 0.962 | 0.962 | 0.971 |
| AP_Omentum_Ovary_115 | 0.975 | 0.956 | 0.956 | 0.975 | 0.944 | 0.944 | 0.862 | 0.644 |
| AP_Omentum_Uterus_11 | 1.000 | 1.000 | 1.000 | 1.000 | 1.000 | 1.000 | 0.962 | 1.000 |
| OVA_Colon_1161 | 0.921 | 0.918 | 0.918 | 0.941 | 0.948 | 0.932 | 0.912 | 0.932 |
| OVA_Omentum_1139 | 0.991 | 0.986 | 0.986 | 0.982 | 0.960 | 0.975 | 0.711 | 0.944 |
| madelon_1485 | 0.949 | 0.941 | 0.942 | 0.907 | 0.845 | 0.891 | 0.607 | 0.625 |

*Table 13.* Healthcare (Bischl et al., 2021) Test Results (AUC ↑).

| Dataset | TabPFN$_{n.e.}$ (RAD-TFM) | TabPFN$_{n.e.}$ (Base) | TabPFN$_{n.e.}$ (Finetuned) | CatBoost | RF | XGB | LR | MLP |
|---|---|---|---|---|---|---|---|---|
| SPECTF_1600 | 0.844 | 0.852 | 0.852 | 0.844 | 0.825 | 0.809 | 0.780 | 0.728 |
| SPECTF_337 | 0.949 | 0.935 | 0.936 | 0.950 | 0.951 | 0.924 | 0.928 | 0.717 |
| SPECT_336 | 0.865 | 0.862 | 0.862 | 0.854 | 0.828 | 0.831 | 0.847 | 0.767 |
| abalone_720 | 0.905 | 0.905 | 0.905 | 0.892 | 0.889 | 0.880 | 0.888 | 0.838 |
| audiology_999 | 0.990 | 1.000 | 1.000 | 1.000 | 1.000 | 0.971 | 1.000 | 0.902 |
| blood-transfusion-se | 0.775 | 0.771 | 0.771 | 0.762 | 0.773 | 0.729 | 0.778 | 0.733 |
| bodyfat_778 | 1.000 | 1.000 | 1.000 | 1.000 | 1.000 | 1.000 | 0.947 | 0.858 |
| breast-cancer-droppe | 0.538 | 0.532 | 0.532 | 0.450 | 0.421 | 0.404 | 0.503 | 0.456 |
| breast-cancer_13 | 0.696 | 0.655 | 0.655 | 0.619 | 0.595 | 0.655 | 0.613 | 0.589 |
| cardiotocography_146 | 1.000 | 1.000 | 1.000 | 1.000 | 1.000 | 1.000 | 0.978 | 1.000 |
| cardiotocography_156 | 1.000 | 1.000 | 1.000 | 1.000 | 1.000 | 1.000 | 0.975 | 1.000 |
| chatfield_4_820 | 0.993 | 0.986 | 0.986 | 0.979 | 0.975 | 0.907 | 0.986 | 0.479 |
| dermatology_1010 | 1.000 | 1.000 | 1.000 | 1.000 | 1.000 | 1.000 | 1.000 | 1.000 |
| haberman_43 | 0.826 | 0.799 | 0.804 | 0.734 | 0.617 | 0.565 | 0.821 | 0.679 |
| heart-c_982 | 0.924 | 0.933 | 0.933 | 0.916 | 0.891 | 0.887 | 0.920 | 0.882 |
| heart-h_1565 | 0.608 | 0.606 | 0.607 | 0.705 | 0.694 | 0.659 | 0.541 | 0.641 |
| heart-h_963 | 0.856 | 0.847 | 0.847 | 0.837 | 0.792 | 0.775 | 0.813 | 0.804 |
| mfeat-factors_978 | 1.000 | 1.000 | 1.000 | 1.000 | 0.997 | 0.999 | 0.999 | 0.997 |
| mfeat-fourier_971 | 1.000 | 1.000 | 1.000 | 1.000 | 1.000 | 1.000 | 1.000 | 1.000 |
| mfeat-karhunen_1020 | 1.000 | 1.000 | 1.000 | 1.000 | 1.000 | 0.999 | 0.998 | 0.996 |
| mfeat-morphological_ | 1.000 | 1.000 | 1.000 | 1.000 | 1.000 | 1.000 | 1.000 | 1.000 |
| mfeat-pixel_1022 | 1.000 | 1.000 | 1.000 | 1.000 | 1.000 | 1.000 | 0.996 | 0.999 |
| mfeat-zernike_995 | 1.000 | 0.999 | 0.999 | 1.000 | 0.999 | 1.000 | 0.972 | 0.968 |
| mu284_880 | 1.000 | 1.000 | 1.000 | 1.000 | 1.000 | 1.000 | 1.000 | 0.738 |
| pbc_810 | 0.753 | 0.757 | 0.757 | 0.778 | 0.765 | 0.831 | 0.705 | 0.703 |
| pbcseq_802 | 0.833 | 0.857 | 0.855 | 0.902 | 0.901 | 0.933 | 0.728 | 0.666 |
| pwLinear_721 | 0.950 | 0.960 | 0.960 | 0.980 | 0.980 | 1.000 | 0.970 | 0.700 |
| thoracic-surgery_150 | 0.768 | 0.711 | 0.711 | 0.746 | 0.789 | 0.879 | 0.704 | 0.750 |
| thoracic_surgery_432 | 0.768 | 0.775 | 0.771 | 0.796 | 0.730 | 0.768 | 0.768 | 0.571 |
| vertebra-column_1523 | 0.944 | 0.933 | 0.933 | 0.947 | 0.949 | 0.900 | 0.902 | 0.783 |
| vertebra-column_1524 | 0.962 | 0.962 | 0.962 | 0.952 | 0.931 | 0.924 | 0.967 | 0.824 |

