# OpenReview forum: "RAD-TFM: Robust and Domain-Adapted Tabular Foundation Models"
_ICML.cc/2026/Workshop/FMSD — FMSD @ ICML 2026 Poster_

### Official Review · Reviewer_8oFp · 2026-05-19
**RAD-TFM: Robust and Domain-Adapted Tabular Foundation Models - Promising robust generator adaptation for tabular foundation models, with strong empirical gains.**

**Rating:** 7
**Confidence:** 4

**Review:**

This paper introduces RAD-TFM, a robust and domain-adapted continued-training framework for tabular foundation models. The key idea is to move beyond fixed synthetic-data priors by learning a distribution over SCM-based synthetic data generators that is both aligned with real-world datasets from a target domain and adversarially biased toward generator regimes where the current TFM underperforms. The authors formalize this using an estimated optimality gap between the TFM and strong classical baselines, then solve a KL-constrained DRO problem over generator parameters. The resulting adversarial distribution reweights generators according to both domain relevance and current model weakness. The method is instantiated on TabPFN V2 and evaluated across general tabular benchmarks and domain-specific OpenML-derived benchmarks.

Overall, I found the paper relevant, well-motivated, and technically interesting. The empirical results are strong, particularly the improvements over TabPFN V2 and standard fine-tuning on TabPertNet, TabArena, PMLB, Engineering, Genomics, and Healthcare benchmarks. The paper is also clearly written, and the connection between synthetic generator design, domain adaptation, and robustness is compelling.

Strengths

- The main idea is well motivated and relevant: synthetic pretraining distributions are important for TFMs, and adapting them toward realistic and difficult datasets is a promising direction. The optimality-gap objective is better justified than simply training on high-loss synthetic tasks, since it focuses on cases where the TFM is worse than strong baselines.

- The empirical results are strong. RAD-TFM improves over TabPFN V2 and fine-tuned TabPFN V2 across multiple benchmarks, including TabPertNet, TabArena, Engineering, PMLB, Genomics, and Healthcare. The paper is also clearly written, and the algorithmic description is easy to follow.

Areas for Improvement

- The main weakness is that the evaluation setting should be clarified. Since the domain-adaptation stage uses real benchmark datasets and label-based learnability features, the method is not purely zero-shot. The authors should clearly state what data is available during adaptation and whether adaptation is done per benchmark, per domain, or per dataset.

- The paper would also benefit from stronger ablations. In particular, it should separately test domain alignment without adversarial reweighting, adversarial reweighting without domain alignment, ordinary synthetic continued training with the same budget, and loss-based sampling instead of optimality-gap sampling.

- Finally, the method is claimed to be model-agnostic, but experiments are only shown for TabPFN V2 classification. A small experiment on another TFM or on regression would strengthen this claim.

Justification of Score

I recommend acceptance. The paper addresses an important problem for tabular foundation models: how to adapt synthetic pretraining distributions to real-world domains and to model weaknesses. The DRO formulation and optimality-gap-based generator reweighting are technically appealing, and the empirical results are strong across several benchmarks. The work is highly relevant to a workshop on foundation models for structured data. However, I would not rate it as a very strong accept because the evaluation setting needs more clarity, the component-level ablations are incomplete, and the claims of model-agnosticism and domain adaptation are broader than what is currently demonstrated. The paper would benefit from more careful discussion of whether the approach is benchmark/domain-adaptive rather than generally zero-shot, and from stronger controls showing that the proposed DRO mechanism is the main driver of the gains.

---

### Official Review · Reviewer_Ues8 · 2026-05-21
**RAD-TFM: Robust and Domain-Adapted Tabular Foundation Models**

**Rating:** 8
**Confidence:** 3

**Review:**

Summary

The paper introduces a model-agnostic adversarial training framework that enhances Tabular Foundation Models (TFMs) by synthetic data generation. The data generation is done in an adversarial manner, while defining a new term, optimality gap, to keep the generation outcome valuable.  By combining distributionally robust optimization with structural causal models, the framework forces the generator to focus on under-performance gaps. Consequently, the output aligns with real-world target domains instead of relying on fixed priors. When applied to TabPFN V2, the proposed method improves the AUC by up to 11%, when using an additional small (less than 0.1% ) training sample of the original pre-training scale.

Strengths

The strengths of this work begin with its innovative synthetic data generation pipeline. The framework creates diverse datasets by randomizing architectural parameters, such as the number of layers, hidden dimensions, and activation functions, within a Multi-Layer Perceptron (MLP). Building on this foundation, the adversarial training approach is highly elegant. It systematically challenges the foundation model with hard, realistic examples explicitly designed to improve generalization. Furthermore, by formulating this adversarial training as a distributionally robust optimization problem constrained by a K-L ball, the framework prevents overfitting to any single generator parameterization. This formulation also effectively enables domain adaptation. Finally, the practical value of the  method is validated by the empirical evaluation. Experimental results consistently highlight the benefits of the proposed approach when compared against several competitive scenarios.

Areas for Improvement

The framework relies heavily on approximating the true Bayes-optimal loss. In practice, if the baseline estimators underperform or fail to capture specific data distributions, the empirical optimality gap may become artificially inflated. The adversarial generator could then exploit these miscalculated regions, focusing training on poorly approximated spaces rather than genuine model weaknesses.
Additionally, the framework introduces a hyperparameter sensitivity risk. Exploring  how sensitive the optimization is to the choice of the KL radius parameter $\rho$ is key. Furthermore, using the Kullback-Leibler (KL) divergence as the distance metric treats the parameter space purely as a statistical distribution, failing to account for the actual semantic or functional distances between different data-generating mechanisms.
Finally, a practical concern is the computational overhead required to calculate the empirical optimality gap. This process demands continuously running and optimizing a bunch of machine learning methods across thousands of generated datasets.

Detailed Comments

Methodology:
-	Discuss the risk of badly approximating the true Bayes-optimal loss. Is this addressed in the literature?

-	Discuss why Kullback-Leibler sufficient, even though it does not account for semantic differences.

-	Discuss limitation in the conclusions part.

Presentation:

-	Add the full name of all acronyms the first time they are presented.
-	RAD-TFM is defined as a two-stage pipeline. As a reader it was not clear what stage 1 and stage 2 actually is.
-	Add "summary of contributions" to the end of the Intro. This will make an easier transition to the next section.
-	Add a reference to Figures 1 and 2 and to the main result tables from the main text, not just from the appendix. I think that Fig. 1 is not referenced at all.

Justification of Score
The main novelty of this work is the replacement of fixed pretraining priors with an active adversarial synthetic data generator. This generator continuously targets the blind spots of a Tabular Foundation Model. A key strength of the approach is that each stage is designed to address the limitations of the previous one, creating a data-efficient pipeline that pushes the adversary to generate datasets that are both challenging and realistic.
The work addressed the core topics of the workshop and paves a way for further research in the same direction.

---

### Official Review · Reviewer_L7Ds · 2026-05-22
**The manuscript introduces RAD-TFM, a robust and domain-adapted training framework for tabular foundation models. The method adapts synthetic data generators toward realistic target domains and datasets where the current model underperforms. The main contribution is its DRO-based generator-selection strategy, which combines adversarial robustness with domain alignment to improve TabPFN performance.**

**Rating:** 7
**Confidence:** 4

**Review:**

This manuscript introduces **RAD-TFM**, a robust and domain-adapted training framework for tabular foundation models. The main idea is to adapt the synthetic data generator distribution toward realistic target domains and toward datasets where the current tabular foundation model underperforms. The method defines an estimated optimality gap by comparing the model's loss against strong baseline estimators. The authors then formulate generator selection as a distributionally robust optimization problem, using a KL-constrained adversarial distribution over synthetic data generators. This leads to a reweighting strategy of the form

\\[
q_i^\\ast \\propto p_i \\exp\\left(\\eta \\widehat{\\delta}_{\\theta_i}(W)\\right),
\\]

where generators receive higher weight when they are both domain-relevant and associated with larger estimated model underperformance.

The overall quality of the manuscript is solid. The technical formulation is reasonable and well motivated, especially the use of the optimality-gap objective and the distributionally robust optimization framework for generator reweighting. The manuscript addresses an important problem in tabular foundation models: fixed synthetic priors may not adequately cover realistic or challenging regions of the tabular data space.

The clarity of the manuscript is generally good. The motivation, problem setup, and main algorithmic steps are presented in a coherent way. The appendices provide useful details on meta-feature construction, SCM parameterization, algorithms, compute resources, and extended benchmark results. However, the experimental protocol could be described more clearly, particularly how the domain-adaptation reference distribution is constructed and what information is used during adaptation.

The manuscript has a moderately original perspective. Rather than relying only on fixed synthetic priors or standard fine-tuning, the manuscript proposes an adaptive synthetic-data generation strategy that considers both domain relevance and model underperformance. This is a useful perspective for tabular foundation models, given that synthetic data generation plays an important role in their pretraining and continued training.

The significance of the work is moderate. The reported results indicate that RAD-TFM can improve TabPFN V2 over the original model and standard fine-tuning on several benchmarks. The manuscript is relevant to the workshop theme and provides a useful contribution to discussions on robustness and domain adaptation for structured-data foundation models. However, stronger ablations and clearer experimental framing would be needed to fully support the empirical conclusions.

**Pros:**

- The DRO-based formulation is technically interesting and well motivated.
- The estimated optimality-gap objective is more informative than simply maximizing model loss.
- The domain-alignment strategy provides a useful way to connect synthetic generation with real-world tabular datasets.
- The empirical evaluation includes several benchmarks and strong traditional baselines.

**Cons:**

- The domain-adaptation protocol should be clarified, especially regarding what real datasets and labels are used to construct the reference distribution.
- The sensitivity of the method to the baseline models used in the optimality-gap estimator is not fully explored.
- The statistical significance results should be presented more carefully, particularly for benchmarks where improvements are not uniformly significant.
- The computational cost and limitations for regression tasks, larger datasets, and high-cardinality categorical features should be discussed more explicitly.